# Babel-ImageNet: Massively Multilingual Evaluation of Vision-and-Language Representations

## Abstract

Vision-and-language (VL) models with separate encoders for each modality (e.g., CLIP) have become the go-to models for zero-shot image classification and image-text retrieval. The bulk of the evaluation of these models is, however, performed with English text only: the costly creation of language-specific image-caption datasets has limited multilingual VL benchmarks to a handful of high-resource languages. In this work, we introduce Babel-ImageNet, a massively multilingual benchmark that offers (partial) translations of 1000 ImageNet labels to 92 languages, built without resorting to machine translation (MT) or requiring manual annotation. We instead automatically obtain reliable translations of ImageNet concepts by linking them – via shared WordNet synsets – to BabelNet, a massively multilingual lexico-semantic network. We evaluate 8 different publicly available multilingual CLIP models on zero-shot image classification (ZS-IC) for each of the 92 Babel-ImageNet languages, demonstrating a significant gap between English ImageNet performance and that of high-resource languages (e.g., German or Chinese), and an even bigger gap for low-resource languages (e.g., Sinhala or Lao). Crucially, we show that the models' ZS-IC performance on Babel-ImageNet highly correlates with their performance in image-text retrieval, validating that Babel-ImageNet is suitable for estimating the quality of the multilingual VL representation spaces for the vast majority of languages that lack gold image-text data. Finally, we show that the performance of multilingual CLIP for low-resource languages can be drastically improved via cheap, parameter-efficient language-specific training. We make our code and data publicly available: `https://anonymous.4open.science/r/Babel-ImageNet-EDBB`

## 1 Introduction

CLIP models (Radford et al., 2021; Jia et al., 2021; Pham et al., 2021) have arguably become the most widely used vision-and-language (VL) models, owing popularity to efficient inference based on separate yet semantically aligned encoders for the two modalities. Their bi-encoder architecture makes them ideal for efficient image-text retrieval (Lin et al., 2014; Plummer et al., 2015) and zero-shot image classification (Radford et al., 2021). They can also produce input vectors for supervised tasks such as image generation (Rombach et al., 2022) or cross-modal reasoning (Eichenberg et al., 2022; Li et al., 2023).

Motivated by the observation that performance on ImageNet classification translates well to performance in many other image tasks (Recht et al., 2019; Fang et al., 2023), CLIP models are typically evaluated on zero-shot image classification (ZS-IC), i.e., by comparing the representation of an image with text representations of class labels, whereby ImageNet (Deng et al., 2009) is the most prominent benchmark. With ImageNet class labels available only in English, this supports only evaluation of monolingual English models (i.e., models trained with English captions only). Although most CLIP models are trained on English-only image-caption data, some effort has been put into creating multilingual and monolingual non-English models by (1) training them from scratch (Bianchi et al., 2021; Ilharco et al., 2021; Yang et al., 2022; Jain et al., 2021) or (2) distilling them from English models (Carlsson et al., 2022; Chen et al., 2022; Zhang et al., 2022), typically using parallel data as supervision. Despite attempts to translate ImageNet labels to other languages (Bianchi et al., 2021;

Yang et al., 2022), the language coverage remains very limited. Because of this, multilingual CLIP models have mainly been benchmarked on image-text retrieval datasets (Aggarwal & Kale, 2020; Bugliarello et al., 2022, *inter alia*), which predominantly cover only limited sets of mid-to-high resource languages.

**Rationale.** Creating *massively multilingual* gold-standard datasets for VL tasks (e.g., image-text retrieval) is prohibitively expensive. Existing efforts (Aggarwal & Kale, 2020; Bugliarello et al., 2022; Thapliyal et al., 2022) either hire native speakers to write image captions in target languages or resort to machine translation (MT) of English data, followed by manual post-editing by native speakers. The MT approach (the cheaper of the two), is, we argue, still too expensive for low-resource languages because MT models are less accurate when translating to those languages, which implies a bigger post-editing effort for bilingual annotators, native in the low-resource language and fluent in English; in addition, such annotators are more difficult to find for low-resource than for high-resource target languages (compare, e.g., Sinhala and German). In this work, we thus seek to create a robust *massively multilingual benchmark* for evaluating the quality of representation spaces of multilingual VL models, without resorting to MT or requiring any manual annotation effort. To be useful, such a benchmark needs to satisfy a crucial requirement: models' performance across languages must be indicative of their performance for the same languages in tasks such as image-text retrieval, for which creating massively multilingual (gold-standard) evaluation datasets is too expensive.

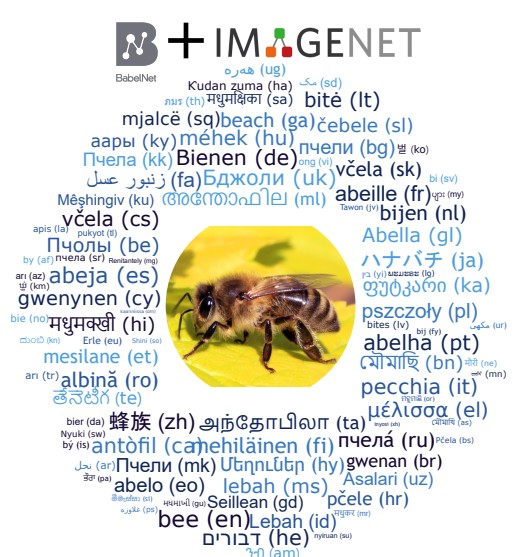

Figure 1: Babel-ImageNet translations in 92 languages for the ImageNet class 309 (synset n02206856: "bee"). Font size of the language is proportional to the number of translated ImageNet classes. [Image by Charles J. Sharp, CC BY-SA 3.0, via Wikimedia Commons]

**Contributions.** With this in mind, we create Babel-ImageNet, a massively multilingual dataset for zero-shot image classification that offers (partial) translations of the 1000 ImageNet-1k classes to 92 languages. To obtain robust translations of ImageNet labels in other languages, we leverage the connection between ImageNet classes, which are derived from WordNet (Miller, 1994) synsets, and BabelNet (Navigli & Ponzetto, 2010), a massively multilingual lexico-semantic network, also (in part) derived from WordNet. Relying on the multilingual BabelNet synsets (and WordNet synset identifiers of ImageNet classes) to pivot between languages we avoid problems known to occur with machine translation of short phrases without context, e.g., due to polysemy[1]. Exploiting BabelNet allows us to automatically obtain labels for ImageNet concepts in many languages, removing the need for MT and manual annotation.

We evaluate 8 different multilingual CLIP models on Babel-ImageNet, observing that all of them exhibit poor performance for low-resource languages. Crucially, we validate that Babel-ImageNet is a meaningful benchmark for measuring the quality of multilingual VL representations by comparing models' performance on Babel-ImageNet against their performance on established multilingual image-text retrieval datasets. Babel-ImageNet thus allows us to evaluate models in languages not covered by those datasets and it additionally expands the retrieval-focused evaluation with the ZS-IC task in languages included in the established datasets. Finally, we propose a computationally efficient approach for improving multilingual CLIP models for low-resource languages. This modular language specialization approach yields large performance gains (>20% for some of the low-resource languages).

---

[1]For example, the ImageNet class *walking stick* refers to the *insect* and not the *inanimate object*.

## 2 RELATED WORK

We first provide an overview of existing benchmarks for evaluating multilingual VL models, followed by a brief overview of multilingual CLIP models, commonly used for efficient image-text retrieval.

### 2.1 MULTILINGUAL VISION-AND-LANGUAGE BENCHMARKS

Early multilingual VL models (Gella et al., 2017; Wehrmann et al., 2019; Kim et al., 2020; Burns et al., 2020; Ni et al., 2021; Geigle et al., 2022; Zhou et al., 2021) were often evaluated in image-text retrieval on Multi30k (Elliott et al., 2016; 2017; Barrault et al., 2018), an extension of Flickr30k (Plummer et al., 2015) to German, French, and Czech, as well as on the Japanese (Yoshikawa et al., 2017) and Chinese (Li et al., 2019) translations of MSCOCO (Lin et al., 2014). More recent models were evaluated on multilingual image-text retrieval benchmarks: XTD (Aggarwal & Kale, 2020) (10 languages) and WIT (Srinivasan et al., 2021) (108 languages). Both these benchmarks, however, have prominent shortcomings. XTD predominantly contains examples from Karpathy's training portion of MSCOCO (Karpathy & Fei-Fei, 2017), which is commonly used for pretraining of VL models: this constitutes a case of data leakage because XTD's multilingual captions are directly translated from the original English captions (Bugliarello et al., 2022). WIT collects image-caption pairs from Wikipedia(s), which leads to two problems: (1) Wikipedia captions are abundant with named entity mentions, which are often identical across a number languages – this artificially equates the difficulty of retrieval across languages (Zhai et al., 2022); (2) The number of image-caption pairs for a language depends on the size of its Wikipedia: this not only prevents direct performance comparisons across languages but also results in overly optimistic performance estimates for low-resource languages with merely a few hundred image-caption test pairs. More recently, IGLUE (Bugliarello et al., 2022) was introduced as the first benchmark to also include reasoning tasks like visual QA (Pfeiffer et al., 2022), primarily meant to test cross-encoder models that jointly encode image-text pairs (Ni et al., 2021; Zhou et al., 2021; Zeng et al., 2022). IGLUE also introduces the image-caption dataset xFlickrCo, a combination of Flickr30k and MSCOCO with captions in 7 languages. A recent dataset, XM3600 (Thapliyal et al., 2022), encompasses 3600 images (balanced by geography of origin) with captions in 36 languages. Despite high-quality[2], it has yet to become widely adopted – along with Zhai et al. (2023), we are among the first to use it.

Motivated by monolingual CLIP models in other languages, translations of ImageNet classes have emerged for a handful of high-resource languages: Italian (obtained with MT) (Bianchi et al., 2021), Chinese (human translations) (Yang et al., 2022), and Japanese[3] and Arabic[4] (undisclosed translation methods). Extending ImageNet to more languages, however, is not feasible at scale: on the one hand, manual translation requires finding native speakers of low-resource languages (additionally fluent in English), which is challenging; machine translation of concepts (i.e., words and short phrases) out of context, on the other hand, is problematic due to polysemy – it thus also requires validation of the translated senses by bilingual annotators: this is especially critical for low-resource languages for which current MT systems are still lacking (Costa-jussà et al., 2022). In contrast, by exploiting the fact that ImageNet classes correspond to Wordnet synsets which, in turn, have corresponding massively multilingual synsets in BabelNet, we are able to create the first robust massively multilingual translation of ImageNet classes, avoiding the caveats of polysemy associated with automatic translation of concepts.

### 2.2 MULTILINGUAL CLIP

While CLIP Radford et al. (2021) is not the first model for embedding images and text in the shared representation space, it has arguably become the most widely used one, owing its effectiveness – especially in ZS-IC – to the immense pretraining corpus. Older models, not exposed to large-scale VL pretraining, e.g., (Faghri et al., 2018) for English and (Gella et al., 2017; Wehrmann et al., 2019; Kim et al., 2020; Burns et al., 2020) multilingually, focused predominantly on text-to-image retrieval and were not shown to exhibit ZS-IC abilities. MURAL (Jain et al., 2021) was the first – albeit not publicly

---

[2]The authors explicitly acknowledge the very high cost of human captioning of images in 36 languages (regretfully, they do not explicitly disclose the cost).

[3]https://github.com/rinnakk/japanese-clip

[4]https://github.com/LAION-AI/CLIP_benchmark/pull/68

released – multilingual CLIP model, trained on billions of multilingual image-caption pairs. To the best of our knowledge, the only publicly available multilingual CLIP models trained "from scratch" are the OpenCLIP models (Ilharco et al., 2021) – pretrained using the full multilingual LAION5B dataset (Schuhmann et al., 2022), consisting of 5B image-caption pairs covering 100+ languages. Monolingual CLIP models for a few languages other than English (e.g., Italian, Chinese) have also been released (Bianchi et al., 2021; Yang et al., 2022): due to comparatively small pretraining data, they trail the English performance. Given the huge computational cost of training a multilingual CLIP from scratch, teacher distillation (Reimers & Gurevych, 2020) has become popular as an efficient alternative (Carlsson et al., 2022; Chen et al., 2022; Zhang et al., 2022): a pretrained multilingual text encoder (e.g., XLM-R Conneau et al. (2020)) is forced (commonly using parallel sentences) to align its representation space to the text encoder of English CLIP.

## 3 BABEL-IMAGENET: MASSIVELY MULTILINGUAL ZERO-SHOT IMAGE CLASSIFICATION

**Why (massively) multilingual ZS-IC?** With class labels in a particular language we can evaluate VL models in language-specific ZS-IC. Note that the goal is not to improve the image classification performance – using labels in any other language yields worse performance compared to using English labels. Instead, we argue that a model's language-specific ZS-IC performance is a good estimate of the quality of its multilingual VL representation space for the language, and thus a good predictor of the model's performance for that language in "real" tasks (e.g., image-caption retrieval).

**WordNet as a matchmaker for ImageNet and BabelNet.** Unlike in most image classification datasets (e.g., CIFAR10, Oxford Pets (Parkhi et al., 2012), Flowers102 (Nilsback & Zisserman, 2008)), where image classes are *words*, ImageNet (Deng et al., 2009) links images to *concepts*, represented with sets of synonyms (synsets) from English WordNet (Miller, 1994). BabelNet (Navigli & Ponzetto, 2010) is a massively multilingual lexico-semantic network, automatically created by merging and consolidating numerous lexico-semantic resources: from WordNets in dozens of languages (e.g., Hamp & Feldweg (1997); Pianta et al. (2002)) to (massively multilingual) Wikipedia and WikiData Vrandečić (2012).[5] Crucially for our efforts, BabelNet is (1) also organized in (multilingual) synsets, containing synonyms across many languages and (2) each of its synsets has an explicit link to the corresponding (English) WordNet synset (if such exists). With WordNet as the seam between ImageNet and BabelNet, we are able to create a massively multilingual ZS-IC benchmark, without resorting to manual annotation or MT.

**Language Selection.** While one could obtain ImageNet class translations in all BabelNet languages,[6] we limit our evaluation to 92 non-English languages (counting unique ISO codes) covered by the pretraining corpora of XLM-R (Conneau et al., 2020). The motivation for this decision is twofold: (1) 50K images in ImageNet and up to 80K label + prompt combinations per language – to be embedded with each model included in our comparative evaluation – make even the ZS-IC evaluation computationally intensive, limiting the total number of languages for which we could carry it out with our computational resources; (2) the majority of publicly available multilingual CLIP models have been derived precisely from XLM-R as the initial multilingual text encoder (see Table 1).

**Class Label Translation and Cleaning Process.** For each ImageNet class, we first fetch all words of each synsets in any of the 92 languages (where available) and English from the corresponding BabelNet synset (using the WordNet synset ID for matching). We next remove words from other languages for which an identical English word exists. We do this because having target language labels identical to English labels would allow multilingual VL models to rely on their high-quality English representations. With those being semantically (much) better than representations of other languages, this would lead to misleadingly optimistic estimates of models' multilingual abilities. On average, this removes $84 \pm 39$ classes from a language-specific benchmark (i.e., we lose classes for which all available BabelNet words in a language also exist as English words in the same synset). Next, we eliminate all words that were added to BabelNet via machine translation, removing the potential negative effects of context-agnostic MT from our benchmark. This mostly affects high(er)-

---

[5]We use BabelNet v5.2, which consolidates 53 sources: `https://babelnet.org/statistics`

[6]BabelNet v5.2 covers 520 languages. We found 298 of them present in at least 10 synsets that correspond to classes of ImageNet-1K; we release labels for these 298 languages with at least 10 classes. We also release our code, so that anyone with access to BabelNet can create translations in additional languages.

resource languages and removes on average $148 \pm 184$ classes. Finally, we select for every remaining class the first words in the respective language (according to the order in BabelNet) as our final language-specific class label. While the above process is not error-free, we estimate[7] that less than 1% of labels are incorrect. We believe this to be a very acceptable error rate, considering (i) that around 6% of ImageNet images are mislabeled (Northcutt et al., 2021) in the first place and (ii) that there are also erroneous mappings between ImageNet images and (English) WordNet synsets (Nielsen, 2018; Radford et al., 2021).

**Grouping Languages in Evaluation.** On the one hand, comparing models' performance over 92 languages (+English) is unwieldy; averaging performance across all languages, on the other hand, is too reductive and consequently not particularly informative. We thus opt for the middle ground: we group Babel-ImageNet languages in three buckets based on their number of classes (less than 333, 334 to 667, and 668 to 1000). We argue that the number of classes is a reasonable proxy for general "resourceness" of a language (see §B.1 for the full list of languages and corresponding numbers of classes) and accordingly designate the three groups as *low-*, *mid-*, and *high-resource*, encompassing 41, 35, and 16 languages, respectively. Additionally, given that ZS-IC becomes easier with fewer classes, averaging results across languages with more comparable numbers of classes (i.e., within each of our groups) makes more sense than averaging them across all languages. Nonetheless, due to differing sets of classes, we caution against direct performance comparisons of results *between* groups or across individual languages. Instead, for any particular language or language group Babel-ImageNet allows for a direct comparison of competing multilingual VL models.

## 4    COMPARATIVE EVALUATION OF CLIP MODELS

**Models.** We briefly describe the public CLIP variants we benchmark on Babel-ImageNet (overview in Table 1). All of them encode images with Vision Transformers (ViT) (Dosovitskiy et al., 2021) albeit of different sizes and with differently sized input patches (e.g., B-32 = `Base` Transformer with $32 \times 32$-pixel patches).

`OpenAI`: The original CLIP model (Radford et al., 2021), trained contrastively on 400M English-only image-caption pairs. The text encoder is trained from scratch (i.e., not initialized with any pretrained weights).

`OpenCLIP`: The OpenCLIP project (Ilharco et al., 2021) aims to replicate the OpenAI models using the public Laion datasets (Schuhmann et al., 2021; 2022). Two multilingual models have been trained on the the multilingual LAION5B dataset: the B-32 model with the text encoder initialized with the weights of XLM-R-Base and the H-14 model is initialized with XLM-R-Large. The B-32 variant is trained with the original contrastive CLIP objective, whereas the H-14 model was trained via locked image tuning (LiT, (Zhai et al., 2022)) in which the pretrained image encoder of the English H-14 OpenCLIP model is frozen and only the parameters of the text encoder are updated.

`SentenceTransformer (ST)`: One of the first distillation-based multilingual CLIP-like models. It was obtained via the distillation approach of Reimers & Gurevych (2020),[8], using over 50M EN-X parallel sentences (X being one of 49 other languages) as supervision. They distill an (distilled, sentence-based) mBERT student (Devlin et al., 2019) from the English OpenAI B-32 teacher.

`M-CLIP`: Multilingual-CLIP (M-CLIP) Carlsson et al. (2022) is another model distilled using mBERT and OpenAI B-32. Unlike ST, they use automatic translations of 3M English image captions (from public image-text datasets) to 69 languages as parallel supervision for the distillation training. Post-publication, the authors additionally released a set of models[9] with XLM-R-Large (instead of mBERT) as the student initialization and (English) OpenAI B-32, L-14, and OpenCLIP B-16+ as teachers, respectively. These models were trained on 7M captions provided by Li et al. (2022), machine-translated to 48 languages. The original English captions were *not* used in training.

`AltCLIP`: This model by Chen et al. (2022) distills an XLM-R-Large student with OpenAI L-14 as teacher, targeting 9 languages and using as training data a mix of machine-translated captions, multi-

---

[7]Based on manual inspection of class labels for a handful of languages that the authors speak.

[8]https://huggingface.co/sentence-transformers/clip-ViT-B-32-multilingual-v1

[9]https://github.com/FreddeFrallan/Multilingual-CLIP/blob/main/larger_mclip.md

Table 1: CLIP variants benchmarked on Babel-ImageNet with the following information: (i) the source (who trained the model), (ii) the text encoder, (iii) the image encoder (all ViT (Dosovitskiy et al., 2021)), (iv) the training objective (CLIP: contrastive training as in Radford et al. (2021), LiT: locked image tuning Zhai et al. (2022), distill: MSE teacher distillation (Reimers & Gurevych, 2020)), (v) the amount of training data (for "distill" the number of caption pairs, for CLIP/LiT the number of image-text pairs), and (vi) the number of languages covered by the training data.

| Source | Text | Image | Objective | #Data | #Langs |
|---|---|---|---|---|---|
| OpenAI | Base | B-32 | CLIP | 400M | 1 |
| OpenCLIP | XLM-R-Base | B-32 | CLIP | 5B | >100 |
| OpenCLIP | XLM-R-Large | H-14 | LiT | 5B | >100 |
| M-CLIP | mBERT | B-32 | distill | 3M | 69 |
| M-CLIP | XLM-R-Large | B-32 | distill | 7M | 48 |
| M-CLIP | XLM-R-Large | B-16+ | distill | 7M | 48 |
| M-CLIP | XLM-R-Large | L-14 | distill | 7M | 48 |
| SentenceTransformer | (distilled) mBERT | B-32 | distill | >50M | 49 |
| AltCLIP | XLM-R-Large | L-14 | distill+LiT | 50M+100M | 9 |

lingual captions sampled from LAION5B, and aligned English-X sentence pairs. After distillation, the authors additionally fine-tune the model via LiT using selected image-text pairs from LAION5B in the 9 target languages.

**Zero-Shot Image Classification Setup.** We adopt the ZS-IC setup of Radford et al. (2021): for an image-label pair, the image embedding is obtained directly from the image encoder; the label is inserted into 80 different prompt templates (from Radford et al. (2021)), each of which is independently embedded by the text encoder – the final label representation is then the mean of prompt embeddings. The class with the label embedding that is most similar to the image embedding (according to cosine similarity) is taken as the prediction; accuracy (top-1) is the evaluation metric.

*Translating Prompts.* We translate the 80 English prompts used by Radford et al. (2021) to our 92 languages using NLLB (Costa-jussà et al., 2022) (model: *nllb-200-distilled-1.3B*; see §B.2 for details). We show (see §C.1) that translated, language-specific prompts lead to better performance compared to using only the class labels or inserting them into the original English prompts. Moreover (see §C.2), we show that translated prompts yield similar performance as human-crafted prompts in *ar*, *it*, *ja*, and *zh*.

**ZS-IC Results.** Table 2 summarizes the results for the low-, mid-, and high-resource language groups, alongside the English performance. The full results for all 92 languages can be found in the Appendix (Table 7).

Table 2: ZS-IC performance on Babel-ImageNet: average results for low-/mid-/high-resource languages and English. **Bold**: best result in each column, both between models with base (B) and large (L/H) image encoders.

| Model | low | mid | high | en |
|---|---|---|---|---|
| OpenAI B-32 | 4.2 | 4.9 | 9.0 | 61.3 |
| OpenCLIP XLMR B-32 | 15.0 | 31.0 | **39.7** | **62.8** |
| M-CLIP XLMR-L B-32 | 25.7 | 32.8 | 33.3 | 42.6 |
| M-CLIP XLMR-L B-16+ | **25.8** | **34.5** | 36.0 | 46.4 |
| M-CLIP mBERT B-32 | 14.8 | 19.3 | 18.9 | 29.2 |
| ST mBERT B-32 | 9.2 | 15.1 | 17.1 | 38.2 |
| M-CLIP XLMR-L L-14 | **28.1** | 37.7 | 39.5 | 51.6 |
| AltCLIP XLMR-L L-14 | 14.2 | 21.1 | 33.6 | 69.9 |
| OpenCLIP XLMR-L H-14 | 19.5 | **41.1** | **52.4** | **77.1** |

Although differing subsets of ImageNet classes across language-specific benchmarks (see §3) prevent direct comparison of numbers between the language groups, these results make it abundantly clear that multilingual CLIP models perform dramatically worse (i) for high-resource languages than for English, and (ii) for low- and mid-resource languages than for high-resource languages. Note that this is *despite* the classification tasks *a priori* being easiest for low-resource languages (fewer than 333 classes) and hardest for English, where models must distinguish between all 1000 classes of ImageNet-1k.

The English ImageNet performance of the models is *not* indicative of their ZS-IC performance for other languages, especially low-resource ones: for example, OpenCLIP XLMR-L H-14 outperforms M-CLIP XLMR-L L-14 by 25 accuracy points on English ImageNet, yet trails it 8.6 points on average for low-resource languages. We believe that this points to the "curse of multilinguality" of the text encoder – namely that, under a fixed model capacity, an improvement of representation quality for some language(s) comes at the expense of representational deterioration for others. This phenomenon

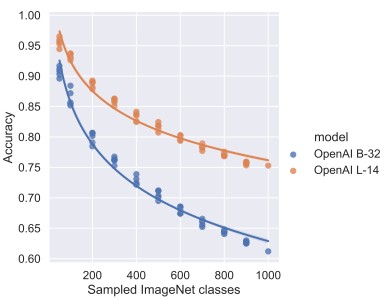

Figure 2: English ImageNet results with random subset of the 1k classes (5 random seeds each).

Figure 3: Multilingual ZS-IC results when considering the same number of classes for each language (we report averages over 5 random subsets of 100 classes per language). For the 12 low-resource Babel-ImageNet languages with <100 classes, we take the single result on all classes.

| Model | low | mid | high | en |
|---|---|---|---|---|
| `OpenAI` B-32 | 5.0 | 7.7 | 15.5 | 84.1 |
| `OpenCLIP` XLMR B-32 | 16.4 | 39.5 | **56.2** | **85.3** |
| `M-CLIP` XLMR-L B-32 | 26.1 | 41.2 | 47.3 | 66.3 |
| `M-CLIP` XLMR-L B-16+ | **26.6** | **42.2** | 48.6 | 70.7 |
| `M-CLIP` mBERT B-32 | 15.9 | 26.5 | 30.3 | 49.9 |
| `ST` mBERT B-32 | 10.4 | 22.3 | 28.6 | 60.1 |
| `M-CLIP` XLMR-L L-14 | 28.2 | 45.2 | 51.9 | 72.2 |
| `AltCLIP` XLMR-L L-14 | 15.7 | 27.5 | 45.7 | 90.9 |
| `OpenCLIP` XLMR-L H-14 | 20.0 | **48.7** | **66.5** | **92.6** |

has been well-documented in particular for XLM-R Conneau et al. (2020); Pfeiffer et al. (2020b). Among the model variants obtained with the same training procedure (e.g., four variants of M-CLIP), English performance does seem to correlate with the performance on other languages.

The OpenCLIP models, trained on the massive LAION5B data, yield good results for high- and mid-resource languages but perform poorly (in comparison with M-CLIP variants) for low-resource languages. In §C.4, we demonstrate that OpenCLIP performance strongly correlates with the distribution of languages in LAION5B: this would suggest that contrastive training (i.e., CLIP and LiT) leads to poor generalization across languages. In contrast, the best performance of M-CLIP models (with XLM-R as text encoder) on low-resource languages suggests that distillation-based training offers better cross-lingual generalization (and yields best performance even for languages unseen in distillation training, see §C.3). We hypothesize that by aligning representations of captions in all other languages to the representations of corresponding English captions results in a more language-agnostic representation space. At the same time, in line with the "curse of multilinguality", this improved generalization is paid with reduced quality of representations of high-resource languages, where M-CLIP models fall well behind OpenCLIP. This trade-off between cross-lingual generalization and per-language performance is best exemplified with AltCLIP: the model is exceptionally good for the 9 languages present in its large-scale distillation training (§C.3), yet performs (comparatively) poorly for most other languages – training on a very large dataset for only a few languages simply overwrites the XLM-R's knowledge of other languages, obtained in its original pretraining.

The two mBERT-based models significantly underperform all other models. This is in part due to mBERT being generally a weaker multilingual text encoder than XLM-R (Hu et al., 2020; Lauscher et al., 2020). On top of that, M-CLIP mBERT variants have been trained on less data than XLM-R-based counterparts (3M vs. 7M captions) and ST is distilled with parallel sentences that are *not* image captions.

## 5 VALIDATING BABEL-IMAGENET

We perform two additional analyses that establish the validity of Babel-ImageNet as a benchmark: (1) how different number of classes affects performance and findings across languages and (2) how multilingual ZS-IC performance on Babel-ImageNet relates to multilingual image-text retrieval performance. We provide further analyses in the Appendix.

**Effect of number of classes on ZS-IC accuracy.** Babel-ImageNet is an incomplete translation of the 1k ImageNet classes (see §3). Intuitively, classification tasks with fewer classes are easier and result in higher absolute performance for all models. We first analyze how the number of classes affects the ZS-IC performance on the English ImageNet-1K, for OpenAI CLIP models (B-32 and L-14). Figure 2 summarizes the results. The task difficulty (i.e., ZS-IC performance) is a log-linear function of the number of classes: this makes intuitive sense – moving from 50 to 100 classes increases the task difficulty much more than going from 900 to 950 classes.

We next fix the number of classes to 100 for all Babel-ImageNet languages (except for languages with <100 classes, for which we make no changes) and report the performance in Table 3 (for each

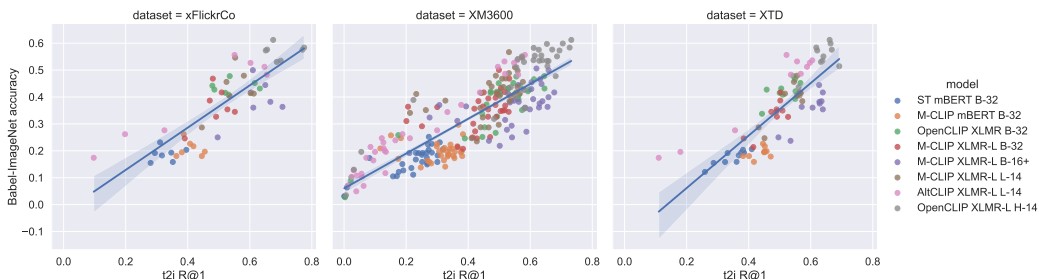

Figure 4: R@1 text-to-image retrieval results on three datasets plotted against Babel-ImageNet performance for 8 CLIP models (each dot denotes the performance of one model for one language) together with a linear regression estimate (95% CI).

language, we average the results over 5 different randomly selected subsets of 100 classes).[10] While in absolute terms the ZS-IC performance increases compared to full class sets (Table 2), and gaps between the language groups widen (especially between mid- and high-resource, and English to all three), our observations do not change: M-CLIP (XLMR-L) still exhibits the best performance for low-resource languages, whereas OpenCLIP is the best for high-resource languages. This renders the (full) Babel-ImageNet a reliable benchmark for directly comparing multilingual VL models.

**Multilingual ZS-IC vs. multilingual image-text retrieval.** The aim of Babel-ImageNet, which measures multilingual ZS-IC performance, is to reflect the quality of the multilingual embedding spaces of VL models. Proving that this is the case implies showing that the models' ZS-IC performance on Babel-ImageNet is indicative of their multilingual performance on tasks that – unlike ZS-IC itself – make sense in a multilingual formulation. The existing body of work commonly evaluates multilingual VL models in image-text retrieval. We thus compare how models' performance on Babel-ImageNet correlates with their performance on three different multilingual image-text retrieval datasets: xFlickrCo (Bugliarello et al., 2022), XTD (Aggarwal & Kale, 2020), and XM3600 (Thapliyal et al., 2022), covering 7, 10, and 36[11] languages, respectively. We use R@1 in text-to-image retrieval as the evaluation metric: it captures the percentage of examples where a correct image is top-ranked for a given caption. We report the full text-to-image retrieval results in the Appendix (Tables 8, 9, and 10).

Figure 4 displays the retrieval results on xFlickrCo, XM3600, and XTD, respectively, against the ZS-IC accuracy on Babel-ImageNet: each dot represents one model-language combination. The plots reveal high correlation between the Babel-ImageNet and text-to-image retrieval scores across model-language pairs: 0.82 for xFlickrCo, 0.87 for XM3600, and 0.85 for XTD. It is particularly positive that Babel-ImageNet shows the highest correlation with XM3600, with which it intersects in most languages (33). Balancing the number of classes to 100 for each language results in even higher correlations, e.g., 0.89 against XM3600. These results confirm that Babel-ImageNet is a sensible benchmark for comparing proficiency of VL models for a multitude of languages not covered in task-specific (e.g., image-text retrieval) benchmarks.

## 6 IMPROVING MULTILINGUAL CLIP FOR LOW-RESOURCE LANGUAGES

Finally, we improve the performance for low-resource languages by resorting to parameter-efficient fine-tuning with adapters (Houlsby et al., 2019; Pfeiffer et al., 2020b), trainable bottleneck layers that we insert into the text encoder. We only update adapter parameters, keeping the original CLIP parameters frozen. We train a separate adapter for each language on top of the same CLIP model, which is computationally much cheaper than fine-tuning a specialized CLIP model for each language.

---

[10]It is not possible to select the exact same set of classes across all Babel-ImageNet languages because only one ImageNet-1k class (*bee*, Figure 1) has BabelNet translations in all 92 languages.

[11]For correlation analysis, we exclude *fil*, *mi* and *quz* as they are not within the 92 languages in Babel-ImageNet; we still report the models' image-text retrieval performance for those languages in Table 9

Table 3: Results of adapter-based language adaptation of M-CLIP and OpenCLIP with three objectives (Text Contrastive, Text MSE, and LiT). Comparison against (i) the model without language adaptation and (ii) best-performing of all 8 CLIP models (see Table 1) for each language. Colors denote the size of change in performance w.r.t. original model: $\leq -5$, $\leq 0$, $\leq 5$, $\leq 10$, $\leq 20$, $> 20$ (best viewed in color). We additionally report language statistics: the number of classes in Babel-ImageNet, the number of tokens used in XLM-R pre-training (in millions, log10), the number of examples in LAION5B (log10) and whether the language was used in M-CLIP training (True/False).

| Model | Loss | xh | si | lo | ur | my | hi | ms | et | sk | lt | eu | ar | ko | fa | de | zh |
|---|---|---|---|---|---|---|---|---|---|---|---|---|---|---|---|---|---|
| Overall Best | No training | 27.8 | 36.3 | 18.0 | 37.2 | 22.4 | 41.1 | 48.5 | 48.1 | 58.8 | 45.7 | 21.4 | 41.7 | 53.2 | 42.6 | 61.2 | 53.5 |
| M-CLIP XLMR-L B-32 | No training | 17.7 | 33.6 | 12.5 | 29.4 | 14.6 | 36.4 | 36.6 | 41.4 | 39.7 | 27.5 | 18.3 | 30.1 | 21.4 | 25.0 | 38.7 | 32.7 |
| | Text Contrastive | 49.0 | 46.1 | 23.4 | 38.4 | 33.3 | 36.2 | 34.0 | 34.6 | 35.8 | 30.5 | 27.1 | 25.3 | 23.4 | 26.1 | 31.4 | 28.8 |
| | LiT | 44.5 | 49.9 | 24.7 | 40.8 | 29.4 | 37.3 | 36.1 | 33.6 | 35.2 | 31.2 | 29.0 | 25.9 | 24.7 | 27.0 | 33.6 | 28.1 |
| | MSE | 46.8 | 53.3 | 26.9 | 43.0 | 37.2 | 42.5 | 39.3 | 38.5 | 41.0 | 35.3 | 34.8 | 29.1 | 29.9 | 31.8 | 38.1 | 31.3 |
| OpenCLIP XLMR B-32 | No training | 24.4 | 3.1 | 0.7 | 25.8 | 5.8 | 25.8 | 37.4 | 29.8 | 45.1 | 35.2 | 17.1 | 24.6 | 33.8 | 32.7 | 47.8 | 40.9 |
| | Text Contrastive | 44.0 | 26.3 | 16.0 | 38.7 | 26.8 | 32.0 | 37.8 | 30.0 | 39.0 | 31.0 | 26.8 | 21.0 | 25.0 | 26.6 | 39.0 | 33.8 |
| | LiT | 47.7 | 33.7 | 19.7 | 37.5 | 23.0 | 30.6 | 35.5 | 28.2 | 38.7 | 30.4 | 28.9 | 22.7 | 27.4 | 27.7 | 39.5 | 30.7 |
| | MSE | 47.6 | 38.7 | 24.7 | 42.8 | 30.4 | 39.0 | 44.0 | 34.6 | 44.2 | 36.7 | 36.2 | 26.9 | 31.9 | 33.0 | 45.9 | 33.6 |
| # Classes | | 35 | 97 | 141 | 220 | 232 | 342 | 419 | 496 | 509 | 535 | 625 | 636 | 648 | 682 | 738 | 885 |
| # XLM-R Tokens | | 1.11 | 2.39 | 1.23 | 2.86 | 1.85 | 3.23 | 3.12 | 2.93 | 3.55 | 3.26 | 2.43 | 3.46 | 3.75 | 4.12 | 4.01 | 2.64 |
| # LAION5B Examples | | 6.71 | 4.11 | 4.07 | 6.13 | 4.49 | 7.18 | 7.05 | 7.01 | 7.06 | 6.98 | 6.73 | 7.35 | 7.01 | 7.32 | 8.18 | 8.16 |
| M-CLIP Distilled | | T | F | F | T | F | T | F | T | F | F | F | T | F | F | T | T |

**Setup.** We train language-specific adapters on top of (a) OpenClip B-32 model (trained from scratch) and (b) M-CLIP XLMR-L B-32 (obtained via distillation). We experiment with three training objectives: English-target language distillation with (i) MSE and (ii) contrastive loss, and (iii) LiT on image-caption pairs. The former two require parallel data, whereas the latter requires images paired with target-language captions. For comparability between languages, we follow Carlsson et al. (2022) and sample 100K captions (with corresponding images) from the synthetic dataset provided by Li et al. (2022) and translate them automatically to all target languages with NLLB. We perform adapter-based specialization for 16 languages. One run (i.e., one model-language-objective combination) takes 3h on a single Nvidia RTX 3090 card (see §B.3 for details).

**Results.** Table 3 displays the results of language adaptation. For low-resource languages (*xh*, *si*, *lo*, *my*, and *eu*), we observe massive improvements, outperforming prior best results by wide margins. In contrast, the adaptation brings performance losses for high-resource languages (e.g., *de* and *zh*). We hypothesize that constraining the representation space of a target language to English representations is beneficial for low-resource languages with semantically poor initial representations, but detrimental for high-resource languages with semantically accurate initial representations. For both OpenCLIP and M-CLIP, adaptation with the MSE objective on parallel sentences yields the best results. Overall, the trends in performance changes from language adaptation are very similar between OpenCLIP and M-CLIP, despite the fact that they were obtained using very different training procedures and trained on datasets with different language distributions. This suggests that this commonality in language adaptation behavior stems from the initialization of the text encoder with XLM-R weights.

# 7 CONCLUSION

We introduced Babel-ImageNet, the first massively multilingual translation of the ImageNet-1k classes to 92 languages. We leverage the WordNet synsets as the link between ImageNet and BabelNet to obtain high-quality translations without relying on MT or human annotators. Using Babel-ImageNet, we carried out the most comprehensively multilingual comparative evaluation of 8 publicly available CLIP models on zero-shot image classification, demonstrating that all models fail for low(er)-resource languages. Crucially, we validate our benchmark by showing that models' text-to-image retrieval performance (on three datasets) strongly correlates with their ZS-IC performance on Babel-ImageNet for the corresponding languages. Finally, we proposed a parameter-efficient fine-tuning procedure that drastically improves the performance of multilingual CLIP models for low-resource languages.

The wide range of languages encompassed by our benchmark reveals that the theoretical "multilinguality" of CLIP models is practically very limited and points to the need for methods that derive robust VL encoders with much stronger performance especially for low-resource languages: e.g., better distillation procedures that retain more of the impressive performance of English CLIP.

## 8 ETHICS STATEMENT

While Babel-ImageNet greatly improves language coverage for the evaluation of multilingual vision-language models, there are some limitations of our work:

For one, the set of classes in ImageNet-1k tend to be Anglo-centric due to inherited biases from WordNet (Shankar et al., 2017; DeVries et al., 2019; Liu et al., 2021) so while our benchmark evaluates the performance on languages from all over the globe, we do not evaluate the model performance on concepts specific (or even unique) to cultures in which the languages are spoken. As a result, Babel-ImageNet may overestimate the actual usability of an VL model for real-world uses in some cultures and geographies.

Further, we select for Babel-ImageNet the 92 languages used in XLM-R pretraining as a tradeoff between language coverage and usability. This selection reinforces research focus on those languages to the detriment of other (mainly extremely low-resource) languages. However, we release our code, as well as data for labels of 298 languages and encourage future research to consider an even wider set of languages.

## 9 REPRODUCIBILITY STATEMENT

All data, code to generate the data (both Babel-ImageNet and MT prompts), and evaluation code can be found at `https://anonymous.4open.science/r/Babel-ImageNet-EDBB`. All datasets and models used in the paper are publicly accessible.

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

Table 4: The 92 languages of Babel-ImageNet in alphabetical order with the corresponding number of classes in Babel-ImageNet.

| af | 303 | am | 85 | ar | 636 | as | 98 | az | 365 | be | 415 | bg | 602 | bn | 282 | br | 297 |
|----|-----|----|-----|----|-----|----|-----|----|-----|----|-----|----|-----|----|-----|----|-----|
| bs | 156 | ca | 767 | cs | 615 | cy | 407 | da | 610 | de | 738 | el | 572 | eo | 603 | es | 845 |
| et | 496 | eu | 625 | fa | 682 | fi | 973 | fr | 799 | fy | 155 | ga | 502 | gd | 217 | gl | 473 |
| gu | 106 | ha | 47 | he | 648 | hi | 342 | hr | 347 | hu | 594 | hy | 454 | id | 463 | is | 409 |
| it | 773 | ja | 733 | jv | 183 | ka | 438 | kk | 365 | km | 167 | kn | 175 | ko | 648 | ku | 101 |
| ky | 247 | la | 276 | lo | 141 | lt | 535 | lv | 392 | mg | 64 | mk | 453 | ml | 281 | mn | 201 |
| mr | 140 | ms | 419 | my | 232 | ne | 134 | nl | 749 | no | 599 | om | 18 | or | 71 | pa | 128 |
| pl | 778 | ps | 112 | pt | 667 | ro | 687 | ru | 748 | sa | 66 | sd | 61 | si | 97 | sk | 509 |
| sl | 393 | so | 58 | sq | 273 | sr | 468 | su | 98 | sv | 699 | sw | 220 | ta | 346 | te | 202 |
| th | 896 | tl | 272 | tr | 559 | ug | 106 | uk | 640 | ur | 220 | uz | 254 | vi | 523 | xh | 35 |
| yi | 175 | zh | 885 | | | | | | | | | | | | | | |

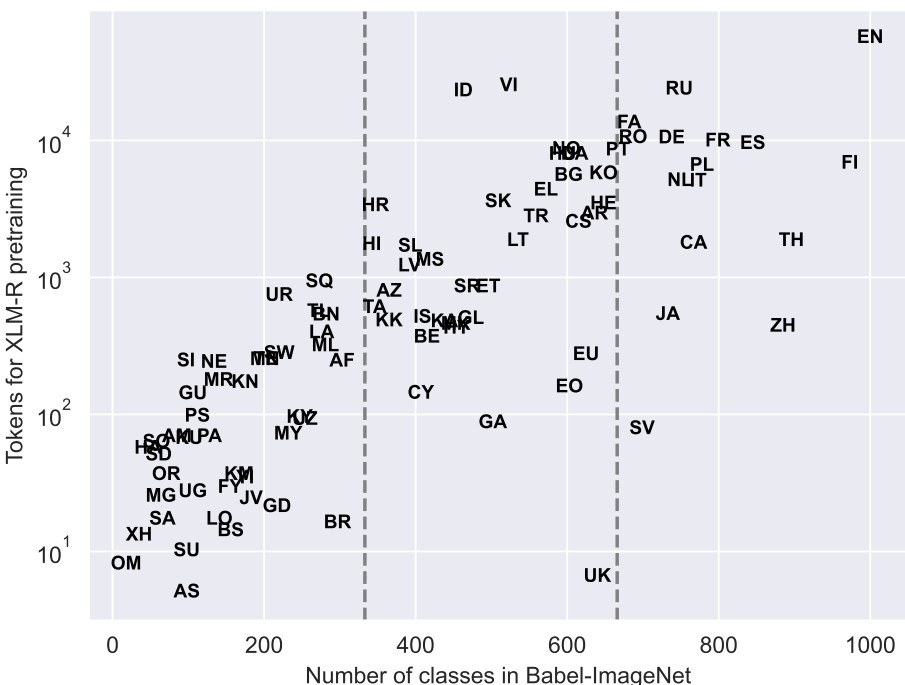

Figure 5: Number of classes in Babel-ImageNet plotted against the number of tokens (millions, log10) in the XLM-R pretraining corpus. When taking the XLM-R tokens as proxy for "resourceness" of a language, we see that this generally correlates with the number of classes. Vertical lines indicate the grouping of languages for evaluation.

## A  LICENSE

Babel-ImageNet is a processed version of BabelNet v5.2 downloaded from `https://babelnet.org`, made available with the BabelNet Non-Commercial License (see `https://babelnet.org/full-license`).

## B  DATA AND TRAINING DETAILS

### B.1  BABEL-IMAGENET

Table 4 lists of the 92 Babel-ImageNet languages with their corresponding number of classes.

Figure 5 visualizes the relationship between the number of classes of a language in Babel-ImageNet and the number of tokens for the language in the XLM-R pretraining corpus (which we use as a

proxy for the language "resourceness"). We see that the two are generally correlated (Spearman rank correlation of 0.78), albeit with some expected outliers, e.g., Chinese is "token-compact" so the token count does not reflect its high-resourceness well.

## B.2 PROMPTS

We use NLLB (Costa-jussà et al., 2022) (*nllb-200-distilled-1.3B*) to translate the 80 prompts used by Radford et al. (2021) to our 92 languages. The exceptions are *fy*, *la*, and *br*, which are not supported by NLLB; for those languages we report the better results between: (1) using only the language-specific labels and (2) inserting 'labels' into English prompts. For the ISO 639-1 languages corresponding to macrolanguages, there is only one corresponding ISO 639-3 language in NLLB, except for *no* where we choose Bokmål and for *az* where we choose North Azerbaijani. We translate the prompts in their template form with the {} placeholders. We use a range of different methods like HTML tags or other special characters to increase the likelihood of preserving the placeholders during translation and then select the first successful approach. If no method worked, we append {} to the end of the sentence. We perform no language-specific adaptions like combining prompt variants with definite and indefinite articles for languages where this distinction does not exist (or articles do not exist at all) nor do we account for the grammatical gender of the classes when inserting them in the template.

## B.3 TRAINING

**Training Data:** For the language-specific adaptation training in §6, we leveraged the BLIP (Li et al., 2022) image-caption dataset CCS_SYNTHETIC_FILTERED_LARGE.JSON[12].

**Hyperparameters:** We train with AdamW (Loshchilov & Hutter, 2019), 0.1 weight decay, a linear learning rate schedule with 20% warmup, learning rate 1e-3 (chosen with sweep over 1e-3, 5e-4, 3e-4, 1e-4), batch size 512 (OpenCLIP)/ 192 (M-CLIP), for 100 epochs (OpenCLIP)/ 15 epochs (M-CLIP; longer training yielded no improvements). Hyperparameters are chosen based on results on Sinhala. We perform no early stopping and use the last epoch for evaluation. The temperature for the contrastive loss is a trainable parameter as in Radford et al. (2021) but we freeze it for text contrastive loss (training it resulted in worse results). The maximum text sequence length is 70. For adapters, we use the Pfeiffer architecture (Pfeiffer et al., 2020b) (task adapters, not language adapters) with reduction factor 16 with the implementation from AdapterHub (Pfeiffer et al., 2020a). We pre-encode images and English captions; i.e. the English embeddings for MSE and contrastive loss are not computed by the trained model but come from the model before training. We do not use any type of image augmentation.

**Negative results:** We experimented with the following methods but did not pursue them further due to not-better or poor results.

1. Training with MSE loss using aligned English-X sentences from WikiMatrix (Schwenk et al., 2021), similar to the ST and (in part) AltCLIP models, resulted in a performance decrease throughout (except for *si* with OpenCLIP) as Table 5 shows. This suggests that it is important to use *"visually-descriptive"* parallel data (i.e., parallel image captions), rather than *any* parallel data.

2. LoRA fine-tuning (Hu et al., 2022) ($\alpha = 8, r = 16$, lr 1e-3 after sweep) significantly ($>10\%$ on *si*) underperformed adapter-based fine-tuning.

3. SimCSE loss (a self-supervised objective) (Gao et al., 2021) based only on target-language captions yielded no improvements compare to the initial model, i.e., without any additional language-specialization training (experimented with OpenCLIP and batch size 256).

4. Multitask training with both LiT and MSE distillation objectives produced no gains compared to training only with the MSE objective.

---

[12]https://github.com/salesforce/BLIP#pre-training-datasets-download

Table 5: Results of adapter-based language adaptation of M-CLIP and OpenCLIP with TextMSE loss using aligned sentences from WikiMatrix. Colors denote the size of change in performance w.r.t. original CLIP model: $\leq -5$, $\leq 0$, $\leq 5$, $\leq 10$, $\leq 20$, $> 20$ (best viewed in color).

| Model | Loss | xh | si | lo | ur | my | hi | ms | et | sk | lt | eu | ar | ko | fa | de | zh |
|---|---|---|---|---|---|---|---|---|---|---|---|---|---|---|---|---|---|
| M-CLIP XLMR-L B-32 | No training | 17.7 | 33.6 | 12.5 | 29.4 | 14.6 | 36.4 | 36.6 | 41.4 | 39.7 | 27.5 | 18.3 | 30.1 | 21.4 | 25.0 | 38.7 | 32.7 |
| | MSE (WikiData) | — | 25.0 | — | — | — | 22.2 | — | 21.3 | 23.3 | 21.8 | 16.9 | 24.0 | 16.3 | 22.5 | 31.4 | 24.3 |
| OpenCLIP XLMR B-32 | No training | 24.4 | 3.1 | 0.7 | 25.8 | 5.8 | 25.8 | 37.4 | 29.8 | 45.1 | 35.2 | 17.1 | 24.6 | 33.8 | 32.7 | 47.8 | 40.9 |
| | MSE (WikiData) | — | 17.1 | — | — | — | 18.1 | — | 18.5 | 25.0 | 23.0 | 17.7 | 21.1 | 14.7 | 23.0 | 38.6 | 28.5 |

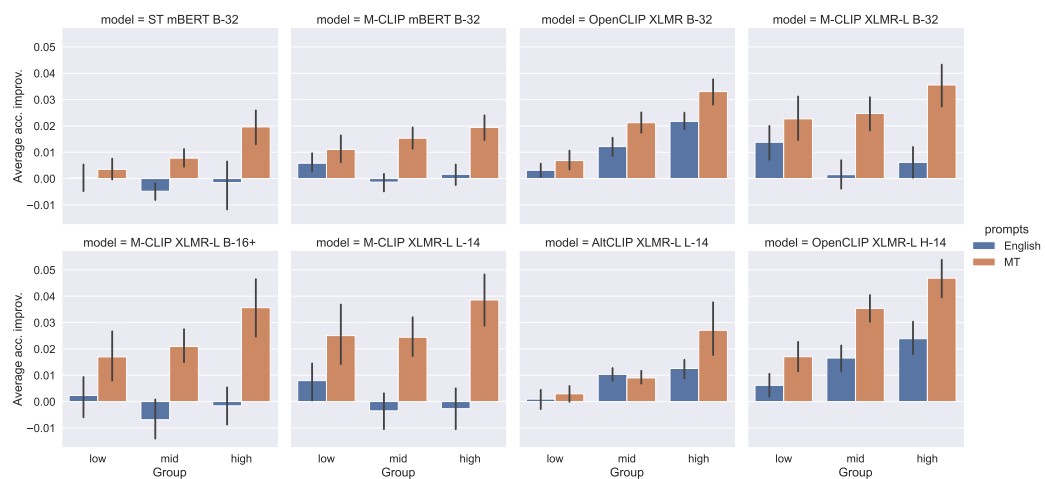

Figure 6: Average increase within the low/mid/high language groups (with 95% CI) over only labels using English prompts (with non-English labels) and our machine-translated prompts.

## C  FURTHER EXPERIMENTS AND ANALYSIS

### C.1  EXPERIMENTAL VALIDATION OF MACHINE-TRANSLATED PROMPTS

We show in Figure 6 that our translated prompts produce better results (on average, across all languages), compared to (i) using just the labels and (ii) inserting the translated labels into the original English prompts created by Radford et al. (2021). With the translated prompts, we get gains of over 2 points for low-resource languages and up to 5 points for high-resource languages.

### C.2  COMPARISON WITH EXISTING IMAGENET TRANSLATIONS

Prior work has created full translations of the 1k ImageNet classes into *ar, zh, jp, it* along with human-written prompts for those languages. We use those translations to validate our BabelNet-derived labels and MT prompts: We evaluate models on the subset of ImageNet classes available for each language in Babel-ImageNet and compare a) only labels and b) human-created templates vs. our MT prompts. Results are shown in Figure 7. While results for *ar* and *it* are slighly higher in absolute numbers on the existing translation, the relative order of models on the Babel-ImageNet benchmarks of those languages is nearly identical to their relative ranking on the respective benchmarks with manually translated ImageNet labels.

We observe that the human-written prompts do not result in a relative improvement over our MT prompts (i.e. no down-shift parallel to the $x = y$ line). In fact, for *it*, our MT prompts even close the gap slightly compared to the label-only setup.

### C.3  PERFORMANCE DIFFERENCES BETWEEN DISTILLED AND NOT-DISTILLED LANGUAGES

With teacher distillation, one would expect the performance in the languages seen in the distillation data to be better than in other languages, not used for distillation. With the wide language selection

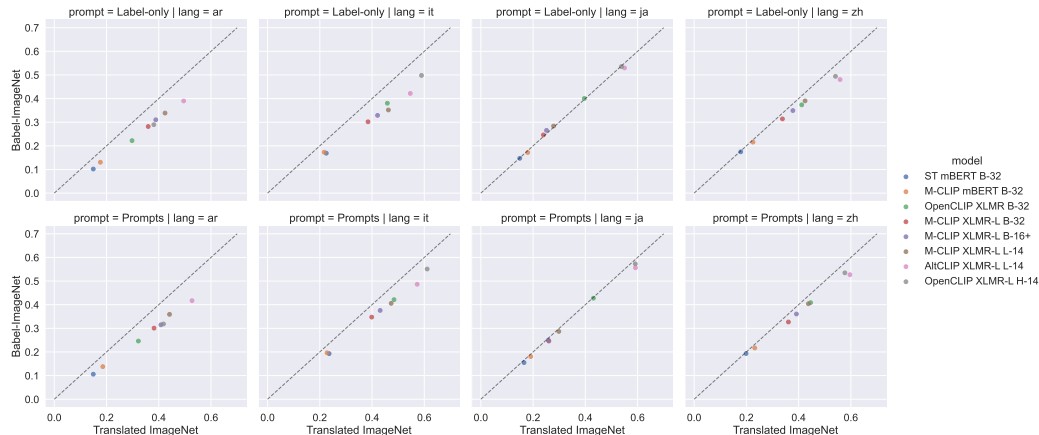

Figure 7: Results on Babel-ImageNet against results on existing ImageNet translations ("labels only" in top row and with our MT prompts vs. human-created prompts in bottom row) for four languages: *ar, zh, jp, it*. Relative model ranking is nearly identical between Babel-ImageNet and manually translated ImageNet benchmarks of respective languages.

Table 6: Average results for the "distilled" languages in the low/mid/high-resource language groups and the $\Delta$ difference to the other "non-distilled" languages of the groups. OpenCLIP H-14 serves as control for language-specific differences in performance not caused by distillation. For M-CLIP, 14/41, 18/35, and 13/16 languages per group are distilled; For AltCLIP, 0/41, 2/35, and 6/16 are distilled.

| **Model** | low | $\Delta$low | mid | $\Delta$mid | high | $\Delta$high |
|---|---|---|---|---|---|---|
| M-CLIP XLMR-L L-14 | 37.6 | +14.4 | 45.9 | +16.8 | 41.6 | +11.3 |
| OpenCLIP XLMR-L H-14 | 26.1 | +10.1 | 47.5 | +13.2 | 55.3 | +15.8 |
| AltCLIP XLMR-L L-14 | — | — | 47.5 | +28.0 | 51.7 | +28.9 |
| OpenCLIP XLMR-L H-14 | — | — | 37.8 | -3.5 | 57.0 | +7.4 |

of our benchmark, we can analyze in-depth how performance on "distilled" languages differs from the performance on "non-distilled" languages.

We compare results for distilled languages on the low/mid/high-resource language groups for M-CLIP and AltCLIP in Table 6; we use the OpenCLIP H-14 model as reference for an expected 'baseline' $\Delta$-difference in performance between the distilled/not-distilled language groups that is due to other factors inherent to the specific languages and not the distillation. For AltCLIP, we see that the the performance on the 8 distilled languages is significantly better than on the non-distilled languages. Moreover, the performance on its distilled languages is even comparable to that of the larger H-14 model. For M-CLIP, the performance on the distilled languages is only slightly better than on the non-distilled low- and mid-resource languages when compared to the OpenCLIP model and the gap is even smaller for high-resource languages. Interestingly, the performance on non-distilled low-resource languages is still noticeably better for M-CLIP than for the OpenCLIP H-14 model. We speculate that the shorter training of M-CLIP compared to OpenCLIP might retain more of the language-specific competences for low-resource languages, obtained in XLM-R pretraining.

## C.4 LAION5B: LANGUAGE DISTRIBUTION AND PERFORMANCE

The distillation-based models evaluated in our benchmark use the same number of training examples for every language. The OpenCLIP models, on the other hand, are trained on LAION5B which follows a more 'natural' distribution of image-caption pairs across languages, as found on the web: Figure 8b shows that over half the data is English, 7 high-resource languages account for another 25% of the data, whereas all remaining languages "share" the remaining 25%.

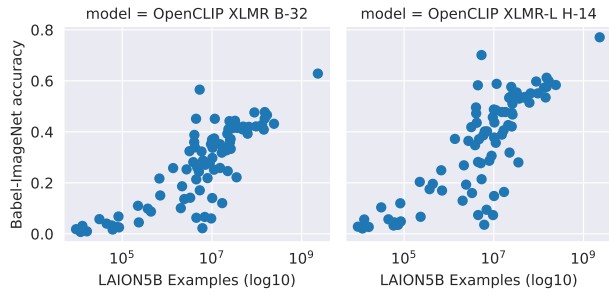
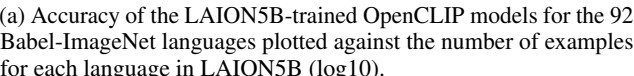
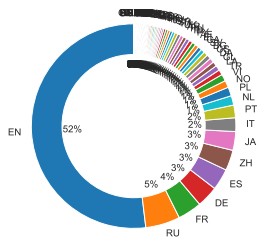

(a) Accuracy of the LAION5B-trained OpenCLIP models for the 92 Babel-ImageNet languages plotted against the number of examples for each language in LAION5B (log10).

(b) Distribution of languages in LAION5B (exluding the 1.3B no-language examples) shows that most examples are either English or one of a few other languages.

Figure 8: The relationship between the LAION5B language distribution with the performance of OpenCLIP models trained on that data.

We can see in Figure 8a that the OpenCLIP's ZS-IC performance for Babel-ImageNet languages highly depends on the number of instances of those languages in the LAION5B dataset. The Spearman rank correlation between the number of language-specific LAION5B examples and the respective Babel-ImageNet accuracy for the language is 0.76. This suggests that pure image-text contrastive pre-training results in poor generalization and limited cross-lingual gains to languages unseen in pretraining. Additional training objectives that aim to better align the multilingual space for example using paired text like in MURAL (Jain et al., 2021) might be necessary to improve results of OpenCLIP-like models (trained from scratch) for low-resource languages.

# D  FULL RESULTS

We report full results for all languages on (i) Babel-ImageNet and (ii) each of the three image-text retrieval datasets: xFlickrCo, XM3600, and XTD.

Table 7: Babel-ImageNet results for all languages (sorted alphabetically expect for English). To save space, we shorten sources (OAI: OpenAI; OC: OpenCLIP; MC: M-CLIP; ST: SentenceTransformer, AC: AltCLIP) and remove the text model if possible.

| lang | OAI B32 | ST B32 | MC mB B32 | OC B32 | MC B32 | MC B16+ | MC L14 | AC L14 | OC H14 |
|------|---------|--------|-----------|--------|--------|---------|--------|--------|--------|
| en | 61.3 | 38.2 | 29.2 | 62.8 | 42.6 | 46.4 | 51.6 | 69.9 | 77.1 |
| af | 10.7 | 14.8 | 24.9 | 36.7 | 46.9 | 49.9 | 55.2 | 22.2 | 47.4 |
| am | 1.5 | 0.9 | 1.4 | 1.0 | 28.8 | 27.1 | 29.0 | 7.6 | 2.7 |
| ar | 0.5 | 10.6 | 13.8 | 24.6 | 30.1 | 31.5 | 35.9 | 41.7 | 31.9 |
| as | 1.0 | 2.9 | 17.5 | 5.4 | 22.6 | 21.4 | 24.4 | 6.5 | 9.8 |
| az | 7.0 | 12.6 | 23.2 | 26.0 | 25.2 | 26.1 | 28.6 | 16.3 | 37.1 |
| be | 0.7 | 9.9 | 14.1 | 27.1 | 32.1 | 34.6 | 36.4 | 27.1 | 40.2 |
| bg | 0.8 | 18.9 | 21.1 | 41.5 | 39.5 | 42.3 | 46.8 | 34.0 | 54.1 |
| bn | 0.2 | 3.0 | 25.7 | 8.7 | 29.1 | 31.1 | 35.1 | 7.1 | 19.6 |
| br | 7.8 | 7.5 | 7.8 | 12.8 | 12.0 | 11.8 | 13.2 | 12.3 | 16.0 |
| bs | 12.9 | 35.3 | 38.9 | 56.6 | 58.9 | 59.5 | 65.2 | 29.4 | 70.1 |
| ca | 11.2 | 17.2 | 19.5 | 33.3 | 33.9 | 36.9 | 39.6 | 31.4 | 46.8 |
| cs | 6.3 | 19.2 | 21.6 | 42.1 | 35.7 | 38.0 | 41.5 | 19.6 | 54.1 |
| cy | 4.9 | 5.5 | 19.4 | 12.0 | 10.9 | 11.1 | 12.6 | 12.6 | 16.4 |
| da | 12.1 | 22.4 | 22.3 | 43.4 | 43.3 | 46.6 | 51.0 | 24.7 | 55.5 |
| de | 15.3 | 18.3 | 19.4 | 47.8 | 38.7 | 41.7 | 45.6 | 27.6 | 61.2 |
| el | 0.9 | 15.1 | 16.3 | 37.2 | 38.5 | 41.2 | 45.2 | 4.9 | 51.0 |
| eo | 5.8 | 10.7 | 12.9 | 21.9 | 24.0 | 24.7 | 28.0 | 24.7 | 28.7 |
| es | 16.6 | 20.4 | 19.7 | 45.2 | 35.4 | 38.2 | 41.4 | 51.2 | 57.6 |
| et | 4.5 | 13.3 | 19.6 | 29.8 | 41.4 | 44.7 | 48.1 | 15.5 | 37.8 |
| eu | 7.5 | 8.7 | 9.5 | 17.1 | 18.3 | 18.7 | 21.0 | 17.6 | 21.4 |
| fa | 0.4 | 13.6 | 20.3 | 32.7 | 25.0 | 27.8 | 28.7 | 20.1 | 42.6 |
| fi | 3.8 | 11.1 | 14.1 | 25.2 | 25.1 | 27.1 | 29.7 | 10.4 | 35.7 |
| fr | 20.7 | 20.6 | 19.8 | 46.6 | 35.0 | 38.4 | 41.7 | 53.6 | 59.9 |
| fy | 12.8 | 17.8 | 17.5 | 35.9 | 35.8 | 35.6 | 37.0 | 25.8 | 42.1 |
| ga | 2.6 | 2.1 | 3.0 | 6.6 | 8.2 | 9.1 | 9.4 | 8.3 | 9.5 |
| gd | 3.0 | 2.7 | 3.4 | 6.0 | 7.3 | 8.4 | 10.4 | 6.1 | 7.4 |
| gl | 15.3 | 23.6 | 22.5 | 44.3 | 39.6 | 40.9 | 44.5 | 43.4 | 54.6 |
| gu | 0.4 | 7.4 | 28.5 | 6.8 | 26.7 | 27.8 | 32.4 | 11.2 | 12.0 |
| ha | 1.5 | 1.8 | 19.8 | 2.2 | 4.5 | 4.6 | 5.3 | 3.3 | 3.6 |
| he | 0.3 | 12.7 | 16.5 | 26.7 | 20.7 | 21.9 | 23.5 | 7.0 | 35.9 |
| hi | 0.1 | 17.3 | 20.9 | 25.8 | 36.4 | 37.0 | 41.1 | 12.8 | 38.9 |
| hr | 8.6 | 25.2 | 29.7 | 45.1 | 50.0 | 50.6 | 56.7 | 21.4 | 58.3 |
| hu | 5.4 | 16.8 | 19.8 | 41.1 | 41.8 | 44.9 | 49.8 | 16.4 | 53.7 |
| hy | 0.2 | 7.3 | 16.5 | 9.9 | 17.6 | 18.2 | 18.2 | 5.3 | 17.5 |
| id | 15.7 | 23.1 | 26.2 | 44.2 | 46.8 | 50.0 | 54.6 | 26.2 | 57.6 |
| is | 2.9 | 6.1 | 18.7 | 13.6 | 39.3 | 41.4 | 45.3 | 10.6 | 19.3 |
| it | 15.1 | 19.3 | 19.6 | 42.1 | 34.7 | 37.6 | 40.6 | 48.6 | 55.1 |
| ja | 4.1 | 15.5 | 18.1 | 42.7 | 24.6 | 25.0 | 28.7 | 55.6 | 57.3 |
| jv | 15.0 | 15.0 | 19.2 | 31.8 | 32.6 | 33.4 | 37.1 | 21.4 | 40.6 |
| ka | 0.2 | 8.5 | 15.1 | 11.0 | 21.6 | 22.5 | 23.8 | 8.9 | 20.4 |
| kk | 0.6 | 10.7 | 19.7 | 28.1 | 25.5 | 26.2 | 27.9 | 31.5 | 34.8 |
| km | 0.9 | 0.7 | 0.6 | 3.3 | 17.5 | 16.5 | 19.3 | 6.7 | 3.3 |
| kn | 1.0 | 2.1 | 21.1 | 4.0 | 27.2 | 27.2 | 29.2 | 10.8 | 5.7 |
| ko | 0.4 | 12.2 | 15.9 | 33.8 | 21.4 | 23.7 | 24.7 | 53.2 | 43.7 |
| ku | 5.1 | 8.9 | 9.4 | 14.2 | 14.1 | 12.5 | 14.6 | 11.2 | 16.0 |
| ky | 0.6 | 13.9 | 18.3 | 32.3 | 28.3 | 28.3 | 31.4 | 32.1 | 38.6 |
| la | 12.3 | 12.5 | 10.5 | 22.2 | 20.2 | 19.8 | 22.2 | 23.9 | 28.0 |
| lo | 0.7 | 0.7 | 0.7 | 0.7 | 12.5 | 12.4 | 13.0 | 9.9 | 2.1 |
| lt | 5.9 | 18.2 | 21.4 | 35.2 | 27.5 | 29.4 | 31.4 | 18.5 | 45.7 |
| lv | 7.8 | 23.4 | 25.7 | 38.8 | 33.5 | 33.4 | 36.7 | 21.3 | 47.2 |
| mg | 4.1 | 9.0 | 8.6 | 14.0 | 14.2 | 13.0 | 14.0 | 13.5 | 14.8 |
| mk | 1.3 | 20.0 | 23.3 | 36.0 | 45.6 | 47.2 | 50.9 | 31.1 | 49.6 |

| | | | | | | | | | |
|---|---|---|---|---|---|---|---|---|---|
| ml | 0.2 | 1.3 | 19.9 | 1.7 | 38.3 | 38.3 | 43.5 | 7.6 | 4.5 |
| mn | 1.0 | 13.9 | 28.0 | 18.7 | 19.4 | 20.9 | 21.5 | 20.2 | 26.9 |
| mr | 1.2 | 29.2 | 25.7 | 32.4 | 45.1 | 45.2 | 47.5 | 17.4 | 41.9 |
| ms | 14.4 | 18.8 | 21.6 | 37.4 | 36.6 | 38.6 | 43.1 | 20.9 | 48.5 |
| my | 0.5 | 10.2 | 2.4 | 5.8 | 14.6 | 14.5 | 16.0 | 9.4 | 10.4 |
| ne | 0.6 | 16.6 | 17.2 | 25.3 | 39.5 | 37.5 | 39.2 | 16.3 | 36.5 |
| nl | 12.7 | 18.5 | 18.2 | 42.0 | 36.5 | 38.6 | 43.9 | 24.7 | 55.1 |
| no | 9.2 | 19.2 | 20.7 | 42.1 | 39.7 | 43.0 | 46.9 | 23.1 | 53.7 |
| om | 5.0 | 8.9 | 3.8 | 5.7 | 8.8 | 11.9 | 10.0 | 8.2 | 13.3 |
| or | 1.9 | 1.7 | 1.7 | 3.2 | 31.0 | 30.8 | 35.9 | 12.4 | 1.7 |
| pa | 1.5 | 4.0 | 8.0 | 1.8 | 32.2 | 33.7 | 31.5 | 11.5 | 2.8 |
| pl | 7.4 | 15.9 | 17.9 | 39.3 | 32.6 | 35.4 | 38.4 | 19.6 | 51.4 |
| ps | 1.3 | 11.6 | 11.1 | 21.7 | 18.8 | 19.4 | 23.0 | 17.9 | 24.9 |
| pt | 14.1 | 22.6 | 21.1 | 47.6 | 39.5 | 42.8 | 46.8 | 37.1 | 59.7 |
| ro | 9.9 | 15.9 | 18.5 | 35.2 | 36.8 | 39.9 | 43.7 | 23.9 | 47.8 |
| ru | 0.6 | 15.8 | 18.0 | 43.2 | 34.5 | 36.3 | 41.3 | 48.3 | 58.4 |
| sa | 0.6 | 9.5 | 9.4 | 10.9 | 17.0 | 17.3 | 18.8 | 11.0 | 11.5 |
| sd | 1.7 | 8.6 | 20.8 | 15.0 | 27.9 | 27.6 | 29.6 | 16.0 | 17.0 |
| si | 2.7 | 2.1 | 2.2 | 3.1 | 33.6 | 33.4 | 36.2 | 18.7 | 5.6 |
| sk | 8.5 | 22.4 | 24.9 | 45.1 | 39.7 | 42.5 | 45.5 | 22.7 | 58.8 |
| sl | 5.9 | 22.0 | 24.2 | 36.9 | 42.3 | 44.4 | 48.4 | 17.2 | 48.8 |
| so | 2.4 | 5.0 | 17.6 | 6.3 | 12.7 | 10.1 | 12.4 | 3.9 | 7.4 |
| sq | 8.1 | 24.3 | 25.5 | 33.9 | 47.1 | 49.3 | 53.6 | 21.6 | 43.6 |
| sr | 1.2 | 17.1 | 18.9 | 33.8 | 43.7 | 45.5 | 49.5 | 30.4 | 47.8 |
| su | 12.1 | 11.6 | 15.9 | 27.9 | 29.2 | 27.8 | 29.2 | 18.6 | 30.6 |
| sv | 8.9 | 19.2 | 20.6 | 43.1 | 41.2 | 45.5 | 48.9 | 20.9 | 55.3 |
| sw | 4.7 | 6.4 | 17.0 | 10.1 | 36.2 | 36.5 | 38.7 | 10.3 | 13.0 |
| ta | 0.3 | 2.2 | 18.4 | 4.5 | 18.0 | 18.7 | 20.6 | 5.5 | 6.7 |
| te | 0.5 | 3.1 | 24.7 | 2.7 | 31.0 | 32.4 | 34.1 | 10.8 | 3.5 |
| th | 1.3 | 10.5 | 15.3 | 28.7 | 27.2 | 29.3 | 32.4 | 11.7 | 40.2 |
| tl | 7.8 | 8.2 | 15.8 | 17.6 | 31.9 | 33.0 | 37.6 | 18.0 | 21.8 |
| tr | 7.5 | 19.2 | 22.6 | 41.5 | 41.7 | 44.5 | 47.9 | 17.4 | 53.0 |
| ug | 1.3 | 2.3 | 2.1 | 3.2 | 11.5 | 12.7 | 12.5 | 9.0 | 4.8 |
| uk | 0.6 | 14.9 | 17.3 | 39.2 | 35.1 | 36.1 | 40.5 | 34.4 | 53.4 |
| ur | 0.6 | 17.5 | 26.8 | 25.8 | 29.4 | 26.9 | 30.8 | 18.6 | 37.2 |
| uz | 6.2 | 9.7 | 21.5 | 21.4 | 22.0 | 21.7 | 23.7 | 17.1 | 28.2 |
| vi | 6.8 | 17.5 | 18.7 | 41.0 | 37.0 | 39.2 | 43.7 | 11.5 | 53.4 |
| xh | 18.7 | 14.2 | 15.8 | 24.4 | 17.7 | 19.1 | 20.2 | 22.2 | 27.7 |
| yi | 0.6 | 1.3 | 1.0 | 2.5 | 18.7 | 18.1 | 19.3 | 5.3 | 4.9 |
| zh | 1.8 | 19.3 | 21.7 | 40.9 | 32.7 | 36.0 | 40.4 | 52.7 | 53.5 |

Table 8: xFlickrCo T2I R@1. Average is without English.

| model | en | de | es | id | ja | ru | tr | zh | average |
|---|---|---|---|---|---|---|---|---|---|
| OpenAI B32 | 51.8 | 10.7 | 20.9 | 4.1 | 1.7 | 0.5 | 2.1 | 0.5 | 5.8 |
| OpenCLIP XLMR B32 | 62.9 | 53.6 | 61.6 | 48.0 | 48.6 | 63.3 | 52.2 | 53.2 | 54.3 |
| OpenCLIP XLMR H14 | 73.9 | 67.6 | 77.0 | 65.6 | 65.2 | 77.6 | 68.7 | 70.0 | 70.2 |
| M-CLIP XLMR B32 | 53.3 | 47.1 | 52.8 | 48.1 | 39.1 | 55.2 | 51.0 | 49.2 | 48.9 |
| M-CLIP XLMR B16+ | 65.3 | 60.4 | 66.7 | 61.0 | 49.7 | 70.6 | 65.0 | 61.2 | 62.1 |
| M-CLIP XLMR L14 | 60.6 | 52.5 | 60.3 | 55.3 | 44.6 | 61.3 | 58.1 | 53.5 | 55.1 |
| AltCLIP XLMR L14 | 64.6 | 32.5 | 64.8 | 19.8 | 55.1 | 65.5 | 9.7 | 60.3 | 43.9 |
| M-CLIP mBERT B32 | 47.0 | 38.1 | 45.5 | 38.7 | 36.4 | 44.6 | 40.8 | 41.9 | 40.8 |
| ST mBERT B32 | 46.3 | 31.7 | 39.6 | 31.1 | 28.1 | 35.2 | 29.4 | 34.7 | 32.8 |

Table 9: XM3600 T2I R@1 results. Average is without English.

| lang | OAI B32 | OC B32 | OC H14 | MC B32 | MC B16+ | MC L14 | AC L14 | MC mB B32 | ST B32 |
|---|---|---|---|---|---|---|---|---|---|
| en | 40.4 | 49.9 | 55.5 | 39.6 | 49.5 | 40.9 | 43.9 | 32.4 | 32.7 |
| average | 4.2 | 43.2 | 51.4 | 41.5 | 52.1 | 44.5 | 21.6 | 29.1 | 20.6 |
| ar | 0.2 | 38.9 | 48.3 | 41.6 | 51.9 | 43.9 | 43.8 | 24.7 | 17.0 |
| bn | 0.0 | 2.1 | 5.5 | 25.1 | 36.5 | 26.3 | 1.3 | 15.1 | 0.2 |
| cs | 1.8 | 47.3 | 56.0 | 42.0 | 53.4 | 44.2 | 10.0 | 29.9 | 23.1 |
| da | 3.5 | 53.1 | 63.6 | 50.6 | 63.2 | 55.1 | 12.1 | 34.3 | 27.0 |
| de | 9.8 | 64.5 | 73.2 | 53.8 | 67.8 | 57.6 | 30.0 | 37.6 | 30.1 |
| el | 0.1 | 45.9 | 55.5 | 43.3 | 52.9 | 46.7 | 3.9 | 26.9 | 18.8 |
| es | 15.4 | 54.6 | 62.8 | 46.4 | 56.3 | 49.1 | 49.6 | 33.3 | 27.3 |
| fa | 0.1 | 49.4 | 58.2 | 41.4 | 52.5 | 44.0 | 13.1 | 31.9 | 21.7 |
| fi | 1.6 | 44.6 | 58.1 | 47.1 | 58.8 | 49.8 | 5.7 | 32.6 | 22.3 |
| fil | 3.7 | 7.2 | 9.8 | 36.6 | 47.3 | 40.0 | 7.3 | 22.9 | 3.7 |
| fr | 19.5 | 61.2 | 70.5 | 51.2 | 63.9 | 55.5 | 55.8 | 37.9 | 31.9 |
| he | 0.2 | 48.9 | 58.5 | 38.3 | 50.2 | 42.8 | 6.3 | 32.2 | 19.1 |
| hi | 0.1 | 17.5 | 22.3 | 20.6 | 30.7 | 23.3 | 3.1 | 11.7 | 8.2 |
| hr | 1.7 | 49.8 | 61.1 | 49.9 | 62.0 | 53.1 | 8.5 | 36.8 | 27.3 |
| hu | 1.8 | 50.4 | 63.8 | 51.8 | 63.6 | 55.9 | 9.4 | 30.6 | 23.1 |
| id | 5.7 | 56.7 | 67.1 | 53.2 | 65.9 | 57.4 | 18.8 | 38.3 | 30.8 |
| it | 8.0 | 59.3 | 68.6 | 50.5 | 63.0 | 53.4 | 51.9 | 35.7 | 26.6 |
| ja | 1.7 | 59.7 | 69.4 | 42.8 | 54.1 | 46.6 | 58.3 | 35.3 | 27.0 |
| ko | 0.1 | 44.5 | 53.7 | 36.6 | 46.7 | 41.1 | 51.7 | 27.5 | 17.6 |
| mi | 0.2 | 0.5 | 0.4 | 0.2 | 0.3 | 0.3 | 0.3 | 0.1 | 0.2 |
| nl | 8.0 | 49.3 | 57.3 | 44.8 | 54.8 | 47.6 | 21.2 | 31.2 | 25.1 |
| no | 3.5 | 50.1 | 61.0 | 48.8 | 60.4 | 52.1 | 12.7 | 34.9 | 24.0 |
| pl | 1.7 | 55.8 | 65.5 | 49.0 | 59.9 | 53.3 | 13.5 | 33.3 | 26.8 |
| pt | 10.1 | 55.5 | 65.4 | 46.5 | 57.8 | 51.0 | 37.7 | 32.4 | 28.3 |
| quz | 1.2 | 3.0 | 3.5 | 1.8 | 2.0 | 1.9 | 3.0 | 1.3 | 1.2 |
| ro | 3.4 | 57.4 | 68.2 | 51.9 | 65.9 | 55.9 | 16.5 | 33.6 | 25.2 |
| ru | 0.5 | 64.6 | 72.7 | 53.0 | 65.6 | 56.6 | 56.1 | 38.1 | 30.1 |
| sv | 3.0 | 52.9 | 61.5 | 49.4 | 60.4 | 53.1 | 12.7 | 34.0 | 25.3 |
| sw | 1.1 | 2.3 | 3.2 | 30.0 | 39.3 | 32.6 | 2.5 | 17.5 | 0.9 |
| te | 0.0 | 0.3 | 0.6 | 20.1 | 27.6 | 21.2 | 4.1 | 18.3 | 0.0 |
| th | 1.3 | 45.3 | 55.4 | 42.2 | 54.0 | 44.7 | 13.3 | 28.1 | 15.8 |
| tr | 1.4 | 47.2 | 58.0 | 45.0 | 56.1 | 48.9 | 7.2 | 29.9 | 21.0 |
| uk | 0.2 | 56.2 | 65.9 | 51.8 | 62.8 | 54.9 | 33.3 | 35.6 | 26.8 |
| vi | 0.8 | 54.9 | 65.4 | 48.8 | 60.9 | 51.3 | 5.6 | 33.9 | 26.6 |
| zh | 0.3 | 56.4 | 62.9 | 46.4 | 58.3 | 50.1 | 54.8 | 36.5 | 28.3 |

Table 10: XTD T2I R@1 results. Average is without English.

| model | en | de | es | fr | it | jp | ko | pl | ru | tr | zh | average |
|---|---|---|---|---|---|---|---|---|---|---|---|---|
| OpenAI B32 | 54.6 | 12.9 | 18.8 | 22.1 | 11.5 | 2.1 | 0.2 | 2.7 | 0.7 | 1.6 | 1.2 | 7.4 |
| OpenCLIP XLMR B32 | 63.0 | 54.4 | 56.7 | 54.9 | 55.1 | 45.8 | 44.3 | 55.0 | 50.5 | 50.0 | 51.0 | 51.8 |
| OpenCLIP XLMR H14 | 72.4 | 66.1 | 66.8 | 66.4 | 64.1 | 62.1 | 56.1 | 69.2 | 61.8 | 63.3 | 63.8 | 64.0 |
| M-CLIP XLMR B32 | 54.7 | 50.2 | 49.0 | 50.9 | 49.0 | 38.4 | 41.5 | 52.7 | 47.3 | 50.3 | 49.1 | 47.8 |
| M-CLIP XLMR B16+ | 66.6 | 63.9 | 62.8 | 63.5 | 62.9 | 49.6 | 53.5 | 63.9 | 57.1 | 62.7 | 60.6 | 60.1 |
| M-CLIP XLMR L14 | 59.1 | 54.6 | 56.8 | 56.5 | 56.4 | 43.2 | 45.1 | 57.3 | 51.7 | 55.6 | 55.6 | 53.3 |
| AltCLIP XLMR L14 | 62.9 | 35.6 | 58.8 | 60.5 | 56.8 | 52.2 | 55.6 | 18.0 | 53.6 | 11.0 | 59.7 | 46.2 |
| M-CLIP mBERT B32 | 50.4 | 45.0 | 45.3 | 45.8 | 45.0 | 38.2 | 41.7 | 47.1 | 40.1 | 44.6 | 45.2 | 43.8 |
| ST mBERT B32 | 49.2 | 36.8 | 38.2 | 41.0 | 38.5 | 28.7 | 25.9 | 36.1 | 32.4 | 33.2 | 37.6 | 34.8 |

