# OpenReview forum: "Babel-ImageNet: Massively Multilingual Evaluation of Vision-and-Language Representations"
_ICLR.cc/2024/Conference — Submitted to ICLR 2024_

### Official Review · Reviewer_3yPQ · 2023-10-31

**Soundness:** 2 fair
**Presentation:** 3 good
**Contribution:** 3 good
**Rating:** 6
**Confidence:** 4

**Summary:**

The purpose of this work is to introduce Babel-ImageNet, a multilingual benchmark for vision-and-language (VL) models that is designed to evaluate their performance in zero-shot image classification and image-text retrieval across several languages. Babel-ImageNet provides translations of (up to) 1000 ImageNet labels in 92 languages without relying on machine translation or manual annotation, just on a multilingual knowledge base. The study evaluates several multilingual CLIP models on the proposed benchmark and shows significant performance disparities, with low-resource languages showing (as expected) the greatest performance gap. Additionally, the paper presents an approach for enhancing the performance of multilingual CLIP models in those low-resource languages.

**Strengths:**

- The study goes beyond traditional monolingual evaluation, offering a comprehensive analysis of 8 multilingual CLIP models across 92 languages.
- The paper is well-motivated and, in general, clear enough to follow through;
- It provides a practical and parameter-efficient approach that significantly improves model performance, making multilingual models more relevant and accessible for underrepresented linguistic communities;
- The dataset/benchmark contribution targets a relevant issue (the overall imbalance between high and low-resourced languages);
- The authors already provide the code for reproducibility purposes;

**Weaknesses:**

- BabelNet reliance. This work relies entirely on BabelNet and assumes that the mapping between WordNet and other resources is high quality. However, BabelNet is automated and has a known percentage of error, potentially affecting the label mapping [1];
- Using WordNet synsets for translations may introduce limitations, as not all concepts or words have direct equivalents in WordNet or BabelNet, potentially impacting the completeness of translations for some languages.
- While the paper emphasizes the creation of the benchmark and model evaluation, it could benefit from a deeper analysis of why certain languages perform poorly according to the chosen metrics and explore potential solutions to address these disparities;
- Considering that the paper belongs to the "datasets and benchmarks" area, the methodology employed (mapping from ImageNet to WordNet and then to BabelNet) is expectedly straightforward.  However, I think there's also some weakness in the data cleaning and validation since the obtained multilingual data is used to evaluate models, but those same benchmarks cannot be used to assess the quality of the data itself;
- The paper's process of removing words with identical English counterparts in the class label translation and cleaning may not be fully justified, as there can be legitimate shared words between the English and language-specific vocabulary;

[1] Ten Years of BabelNet: A Survey. Roberto Navigli, Michele Bevilacqua, Simone Conia, Dario Montagnini, Francesco Cecconi. IJCAI 2021

**Questions:**

- There's a missing reference to another text-image dataset produced by *manual* annotation over BabelNet synsets [1]. In general, I'm curious about the possible usage of this dataset as a proxy to evaluate at least the Babel-ImageNet methodology part that performs prompt translation. I'd like to kindly ask the authors what they think about it or if they have alternative ideas toward strengthening data validation, which I think is the strongest weakness in this current version of the manuscript;
- Not a direct weakness, so I'm listing it here: multiple usages of the verb "demonstrate" (e.g., "we demonstrate that OpenCLIP performance strongly correlates with the distribution of languages in LAION5B" ). Personally, given the empirical nature of this work, I wouldn't suggest using such a theoretical-related term;

Given the doubts about the data validation, I'm starting the discussion period leaning toward rejection. However, I'm open to changing my assessment in light of the rebuttal discussions and the reviews of my colleagues.

[1] Fatality Killed the Cat or: BabelPic, a Multimodal Dataset for Non-Concrete Concepts. Agostina Calabrese, Michele Bevilacqua, Roberto Navigli. ACL 2020

---

> ### Author Response · Authors · 2023-11-15
> **Response 1/2**
>
> Thank you for your comprehensive review!
>
> > BabelNet reliance. This work relies entirely on BabelNet and assumes that the mapping between WordNet and other resources is high quality. However, BabelNet is automated and has a known percentage of error, potentially affecting the label mapping [1];
>
>
> Yes, BabelNet is created automatically and thus not not error-free. But quoting [1]  “For example, in BabelNet 5.0 more than 90% of the mapping between Wikipedia pages and WordNet synsets has been manually validated by experts, resulting in an overall mapping precision above 99.5%.” & “Wiktionary entries are now integrated automatically using a BERT-based neural model, finetuned to associate a word sense definition to its correct synset, attaining an F1 score of 92% on a manually annotated test set.” .
>
> This suggests an error rate that is, to us, in an acceptable range (and substantially lower than any machine translation of concepts in isolation would yield). There is a clear trade-off between the range of “translation” errors and the number of supported languages: manual annotation is (virtually) error free, but does not scale to many languages; MT scales to hundreds of languages, but is very error-prone for concept translation; by “trusting” BabelNet, we can greatly increase the number of languages, if we accept the small error rate present in the resource – getting this way close to a “sweet spot” of this trade-off between the error rate and number of supported languages. In sum, Babel-ImageNet offers a fairly reliable evaluation for a large number of languages, without expensive human annotation, and helps in particular with low(er)-resource languages.
>
> The correlation with retrieval datasets (Figure 3) and the near-perfect correlation w.r.t. the four existing translated ImageNet variants (Arabic, Chinese, Japanese, and Italian) in Figure 7 (Appendix) further empirically validate our BabelNet-derived translations.
>
>
> > Using WordNet synsets for translations may introduce limitations, as not all concepts or words have direct equivalents in WordNet or BabelNet, potentially impacting the completeness of translations for some languages.
>
>
> Can you clarify this point? ImageNet itself is constructed from WordNet synsets so all classes map to a WordNet synset and BabelNet is constructed also from WordNet so all WordNet synsets also appear in BabelNet. The availability of translations in any particular language, however, is affected by other factors, e.g., is there a WordNet equivalent in a language, how big is the Wikipedia/Wikidata of that language, etc.
>
> > While the paper emphasizes the creation of the benchmark and model evaluation, it could benefit from a deeper analysis of why certain languages perform poorly according to the chosen metrics and explore potential solutions to address these disparities;
>
> We are somewhat confused by the second point – exploring solutions to address these disparities -- because our Section 6 is wholly dedicated to this (i.e., improving the multilingual CLIP models’ representational quality for low-resource languages). You have highlighted that as one of the strengths of our work (third bullet point in “Strengths”).
>
> Regarding the analysis point, we perform additional analysis in the Appendix to try and understand why models (do not) perform well for different languages: In D.3., we consider the effect of distillation and in D.4., we look into the effect of the training data language distribution for OpenCLIP models.
>
> Unfortunately, the prohibitive computational expense of training models from scratch ourselves limits us to observational analysis of existing models, so we cannot analyze how the choice of (pre)training data, objective, or initial multilingual text encoder, would affect various languages in controlled experiments.
>
>
>
>
> >Considering that the paper belongs to the "datasets and benchmarks" area, the methodology employed (mapping from ImageNet to WordNet and then to BabelNet) is expectedly straightforward.
>
>
> See our general response to all reviewers.
>
>
> > However, I think there's also some weakness in the data cleaning and validation since the obtained multilingual data is used to evaluate models, but those same benchmarks cannot be used to assess the quality of the data itself;
>
>
> We have commented on the reliability of BabelNet in the first point above. Also, we have validated (Section 5) the meaningfulness of Image-BabelNet via the (small) set of languages that also exist in retrieval benchmarks, showing substantial correlation as well as for the four languages for which translations of ImageNet exist (Appendix, Figure 7). See also our reply in relation to BabelPic below.

---

> > ### Author Response · Authors · 2023-11-15
> > **Response 2/2**
> >
> > > The paper's process of removing words with identical English counterparts in the class label translation and cleaning may not be fully justified, as there can be legitimate shared words between the English and language-specific vocabulary;
> >
> >
> > This is an accurate observation – there are such instances, for example, ‘Zebra’ or ‘Axolotl’. However, as we write in §3, what the multilingual models will use in those cases are their more accurate/stronger English representation, diluting the insight into how good the representational quality is for the other language. By keeping such translations, as a consequence, benchmarks for other languages would “reward” the models for their English “competency”. To illustrate (with an extreme example), imagine a language X with 200 classes, 100 of which are identical to the English label. If a model achieves 50% accuracy on language X, is it actually useful for X or is it just correct on the subset of the X’s concept vocabulary that overlaps with the “English” one? By filtering out all English-matching words, we avoid this problem.
> >
> > > There's a missing reference to another text-image dataset produced by manual annotation over BabelNet synsets [1]. In general, I'm curious about the possible usage of this dataset as a proxy to evaluate at least the Babel-ImageNet methodology part that performs prompt translation. I'd like to kindly ask the authors what they think about it or if they have alternative ideas toward strengthening data validation, which I think is the strongest weakness in this current version of the manuscript;
> >
> > Thank you for the reference!
> >
> > Could you clarify what you mean by “the Babel-ImageNet methodology part that performs prompt translation”  because we do not understand how that relates to BabelPic?
> >
> > Regarding BabelPic methodology, they want to automatically verify if an image and their non-concrete concepts (e.g. “sadness”) match and they test if visual QA models can perform this verification using gold-label human-annotated image-concept data. For Babel-ImageNet, the verification of the match between the WordNet synset and image was already performed by humans in the construction of ImageNet itself.
> >
> > > Not a direct weakness, so I'm listing it here: multiple usages of the verb "demonstrate" (e.g., "we demonstrate that OpenCLIP performance strongly correlates with the distribution of languages in LAION5B" ). Personally, given the empirical nature of this work, I wouldn't suggest using such a theoretical-related term;
> >
> > Thank you for the suggestion. We were not aware that “demonstrate” has such a strong theoretical meaning. We were using “demonstrate” simply as a synonym for “show” (and not a synonym for “prove”). We are happy to change this and simply replace instances of “demonstrate” with “show”.

---

> > > ### Comment · Reviewer_3yPQ · 2023-11-22
> > >
> > > First, I'd like to thank the authors for their thorough reply.
> > >
> > > They have addressed all my concerns and those raised by other reviewers. Given their answers and the other reviews, I'm increasing my score to 6.
> > >
> > > A few comments regarding their answer:
> > >
> > > - BabelNet reliance. While that statement holds for English mapping, the quality of the mapping from other languages to English is not of high level at all, especially considering under-resourced languages;
> > >
> > > - WordNet limitation. My perspective on this point is that WordNet is orders of magnitude smaller than BabelNet (in number of synsets: ~100k vs ~20M). While I agree that the mapping to BabelNet from WordNet is straightforward, it doesn't imply that it gains access to the whole BabelNet senses. Plus, the intersection between WordNet synsets and Wikipedia pages is only ~40k, so the obtained annotations/samples are limited by this.

---

> > > > ### Author Response · Authors · 2023-11-22
> > > >
> > > > Thank you very much for your reply. We are happy to hear that we addressed your concerns.
> > > >
> > > > To your comments:
> > > > * BabelNet reliance: We understand the concern but your citation does not support this claim. While not a quantitative evaluation, based on the checks in the languages we speak and some we do not like Xhosa where we used Google image search and translate as aides, we identified few errors so we think "not high level at all" is exaggerated (at least for to the synsets of ImageNet).
> > > >
> > > > * WordNet limitation: We agree with this in general but we want to highlight that we deal specifically with WordNet synsets used in ImageNet, i.e., highly specific concepts like vulcanos, zebras, and German Shepherds. For those synsets, we do not need diverse senses and they also have, we suspect, more often a corresponding Wikipedia page than the average WordNet synset.

---

### Official Review · Reviewer_rcTe · 2023-10-31

**Soundness:** 3 good
**Presentation:** 4 excellent
**Contribution:** 4 excellent
**Rating:** 8
**Confidence:** 4

**Summary:**

Zero-shot image classification and image-text retrieval evaluation primarily focusses on English only. Curation of high quality evaluation datasets in other languages is expensive and time consuming. This paper proposes e Babel-ImageNet, a massively multilingual
benchmark that offers partial translations of 1000 ImageNet labels to 92 languages, built without resorting to machine translation or requiring manual annotation. It leverages the connection between ImageNet classes, which are derived from WordNet synsets, and BabelNet, a massively multilingual lexico-semantic network, also (in part) derived from WordNet.

Babel-ImageNet thus allows us to evaluate models in languages not covered by other evaluation datasets and it additionally expands the retrieval-focused evaluation with the zero-shot image classification task in languages included in the established datasets.

The paper proposes a computationally efficient approach for improving multilingual CLIP models for low-resource languages. This modular language specialization approach yields large performance gains (>20% for some of the low-resource languages).

**Strengths:**

This paper introduces an extensive image-text evaluation benchmark on a large set of languages which motivates research in the largely unexplored multilingual VL representation learning space. Also, the technique is free from any machine translation or similar techniques that can introduce errors in the evaluation data. This makes it more robust and suitable for adoption. This evaluation corpus should be extremely helpful for furthering research in this area.

**Weaknesses:**

None.

**Questions:**

None.

---

> ### Author Response · Authors · 2023-11-15
>
> We thank you for your review and are happy to read that we convinced you of the merits of our work.

---

### Official Review · Reviewer_3yZg · 2023-11-01

**Soundness:** 3 good
**Presentation:** 3 good
**Contribution:** 3 good
**Rating:** 6
**Confidence:** 3

**Summary:**

This paper proposes a robust and machine-translation-free method to create non-English labels for the ImageNet-1k dataset in 92 different languages. When used to evaluate VL models, the new Babel-ImageNet dataset showed score correlated with retrieval performance on multilingual image-text datasets. Finally the paper used the dataset to evaluate models with parameter efficient tuning toward multilingual capability.

**Strengths:**

- The proposed translation method is robust and the claimed error rate from manual inspection is low.
- The translation covers 92 languages, including many medium and low resource languages.
- When evaluating multilingual models, the performance on Babel-ImageNet correlates well with the text to image retrieval performance on multilingual image-text datasets, suggesting the usefulness of this dataset as an alternative evaluation method for multilingual models

**Weaknesses:**

From a significance and usefulness perspective, the unique advantage of this dataset over the multilingual image-text datasets for model evaluation is unclear. It is not surprising that the performance of models on multilingual ImageNet classification is correlated with multilingual text to image retrieval. My concern is that Babel-ImageNet might not be as good as the multilingual image-text datasets as the former contains much less detailed description for the image, and that other image-text datasets support image-to-text retrieval as well for which Babel-ImageNet could not cover.

The section 6 discussion might be a good opportunity to set up such a comparison if the models there could be evaluated on the multilingual image-text datasets as well. If the authors can show that Babel-ImageNet better reflects the model quality improvement, that would make a strong argument.

**Questions:**

Can you show some cases where Babel-ImageNet has wrong non-English labels? Are there any systematic errors?

---

> ### Author Response · Authors · 2023-11-15
>
> Thank you for your review!
> > From a significance and usefulness perspective, the unique advantage of this dataset over the multilingual image-text datasets for model evaluation is unclear. It is not surprising that the performance of models on multilingual ImageNet classification is correlated with multilingual text to image retrieval. My concern is that Babel-ImageNet might not be as good as the multilingual image-text datasets as the former contains much less detailed description for the image, and that other image-text datasets support image-to-text retrieval as well for which Babel-ImageNet could not cover.
>
> The goal of Babel-ImageNet is not to replace existing datasets but rather to complement them and extend evaluation possibilities primarily along the language axis.
>
> Languages: Coverage of established multilingual image-text retrieval datasets is limited with 7 to 36 languages. Babel-ImageNet allows us to evaluate on substantially more (96 in total) languages and the correlation with retrieval on languages where we have both (retrieval and Babel-ImageNet) suggests that this is a sensible approach.
>
> Complementarity: We agree that image captions are more descriptive, but on the other hand, they are also generally less specific: Any breed of dog is in most captions just denoted “dog”, any type of tree is just a “tree”. ImageNet requires a more fine-grained understanding of objects to differentiate different dog breeds, tree species, etc. As a result, image-text retrieval and classification together make for a more comprehensive evaluation of the representations than either alone. Testing on a range of complementary retrieval and classification datasets/tasks is a standard procedure for English models: with Babel-ImageNet we take a step towards the same for multilingual models.
>
> > The section 6 discussion might be a good opportunity to set up such a comparison if the models there could be evaluated on the multilingual image-text datasets as well. If the authors can show that Babel-ImageNet better reflects the model quality improvement, that would make a strong argument.
>
>
> As mentioned above, we are not claiming that Babel-ImageNet is better for evaluating models than retrieval datasets for (the limited set of) languages covered by those datasets. But your suggestion illustrates the problem we are aiming to solve: only 6/16 languages used in §6 are included in XM3600’s 36 languages and none of those 6 is one of the low-resource languages where we see improvements. Yes, we could have limited ourselves to the languages also covered by retrieval datasets but with Babel-ImageNet’s language coverage, we had more options for the analysis to cover diverse language families, scripts, levels of pre-training data and distilled/not-distilled.
>
>
> > Can you show some cases where Babel-ImageNet has wrong non-English labels? Are there any systematic errors?
>
>
> We have not noticed any systematic errors but, while not exactly wrong, sometimes the label for a plant or animal is its scientific name instead of a more common name (e.g., pumpkin -> cucurbita pepo pepo). Filtering out English options that included scientific names removed most such cases, but some instances remained in other languages.
>
> Filtering options can also result in incorrect choices, although rarely. For example, the concept “Golf ball” in BabelNet is erroneously mapped to German to “Physics of the golf ball” (sourced from Wikipedia page redirects) because the more sensible choice “Golfball” was removed because it matched an English name of the concept (and those we removed not to overestimate the performance of models in other languages on the account of their English competencies, as explained in the last paragraph on page 4).

---

### Official Review · Reviewer_SS5H · 2023-11-03

**Soundness:** 2 fair
**Presentation:** 3 good
**Contribution:** 2 fair
**Rating:** 6
**Confidence:** 4

**Summary:**

This paper introduces Babel-ImageNet, a benchmark that translates English ImageNet labels into 92 languages using BabelNet. Furthermore, the paper evaluates eight different publicly available multilingual CLIP models on this benchmark. Experimental results indicate that there is a high correlation between the zero-shot performance of image classification and their performance in image-text retrieval, thereby validating the high quality of Babel-ImageNet.

**Strengths:**

1) The multilingual ImageNet benchmark, which supports 92 languages, serves as an excellent platform for evaluating multilingual CLIP models, particularly for those languages that are under-resourced.
2) The assessment of eight different multilingual CLIP models also provides valuable insights.

**Weaknesses:**

My concern is about the simplicity of the method, which merely translates English ImageNet labels using BabelNet. While the resulting benchmark proves useful, the method's contribution appears to be limited.

**Questions:**

Have you considered using GPT-4/ChatGPT to prompt the model to translate English ImageNet labels? Perhaps combining GPT-4/ChatGPT with WordNet could yield better results.

---

> ### Author Response · Authors · 2023-11-15
>
> Thank you for your review!
> > My concern is about the simplicity of the method, which merely translates English ImageNet labels using BabelNet. While the resulting benchmark proves useful, the method's contribution appears to be limited
>
> See our general response to all reviewers. We do not disagree that our creation method is straightforward and simple, but we see do not perceive this as a weakness.
>
> > Have you considered using GPT-4/ChatGPT to prompt the model to translate English ImageNet labels? Perhaps combining GPT-4/ChatGPT with WordNet could yield better results.
>
> We have not but it is an interesting idea to explore: LLMs are surprisingly good at translations and if we provide them additional context (e.g. via WordNet synonyms or definitions), we could likely reduce the problem of polysemy that designated NMT models struggle with. However, the main problem here is the same as in traditional MT models: performance degrades massively for low-resource languages, which makes human verification necessary again, which is exactly the problem we want to avoid by exploiting BabelNet which is massively multilingual and readily available.

---

### Author Response · Authors · 2023-11-15
**General Response**

We thank the reviewers for your valuable time and feedback. We are happy to hear that you all find Babel-ImageNet to be a useful and helpful resource, especially for low-resource languages [R1, R2, R3, R4], created through a robust and reliable process [R2, R3]. We are also encouraged to read that you find our evaluation of 8 models comprehensive [R4] and with valuable insights [R1].

R1 and R4 raise some concerns that our construction method is too simple and straightforward. We would argue that simple methods (that solve the problem) should be preferred as they are easier to comprehend, validate for soundness, and more reproducible.
Also, while using BabelNet for translations seems straightforward in retrospect, no prior work on multilingual vision-language representations – many of which automatically or manually translated the ImageNet labels, too – thought of leveraging it.

---

### Meta-Review · Area_Chair_yQN4 · 2023-12-15

**Metareview:**

This paper presents Babel-ImageNet, a significant advancement in the field of multilingual vision-and-language models. It establishes a benchmark by translating ImageNet labels into 92 languages using BabelNet, a process that aims to be more effective than machine translation and more scalable than manual annotation. This benchmark is used to evaluate the performance of various multilingual CLIP models, particularly focusing on zero-shot image classification and image-text retrieval across a diverse range of languages. A key finding is the notable performance disparities in low-resource languages, underscoring the challenge of developing equitable multilingual models.

Among the paper's strengths is its innovative approach to creating a multilingual dataset. By leveraging BabelNet, the authors address the critical need for expansive language coverage in vision-and-language research, a domain where many languages are typically underrepresented. This broad coverage makes Babel-ImageNet a valuable resource for evaluating and enhancing the performance of multilingual models, especially in low-resource languages.

The paper also has some limitations. Reviewers pointed out potential mapping errors in using BabelNet, and although the authors provided clarifications, a comprehensive study assessing the accuracy of BabelNet mappings against machine translations is missing (beyond a single error number). Additionally, a key baseline, translating labels to English for zero-shot image classification to benchmark against the best-performing languag, was not included, which is actually essential as translation should be seen as a feature, not a bug.

A major issue, however, is the breach of the anonymity policy due to a GitHub link in the paper leading to a non-anonymized repository. This violation reveals the authors' identities, which is a critical lapse in adhering to the peer review standards.

**Justification For Why Not Higher Score:**

Same as above.

**Justification For Why Not Lower Score:**

N/A

---

### Decision · Program_Chairs · 2024-01-16

Reject